## RESEARCH ARTICLE

# Drak is a potential binding partner of *Drosophila* Filamin

Riku O. Korkiamäki[1,‡], Chandan Thapa[1], Hannah J. Green[1,2,*] and Jari Ylänne[1]

## ABSTRACT

Mechanosensing involves proteins detecting mechanical changes in the cytoskeleton or at cell adhesion sites. These interactions initiate signaling cascades that produce biochemical effects such as post-translational modifications or cytoskeletal rearrangements. Filamin is a ubiquitous mechanosensing protein that binds actin filaments and senses pulling forces within the cytoskeleton. *Drosophila* Filamin (Cheerio) is structurally similar to mammalian Filamin, with roles in egg chamber development, embryo cellularization, and integrity of muscle attachment sites and Z discs in *Drosophila* indirect flight muscles (IFMs). Here, we report a potential novel binding partner of *Drosophila* Filamins: the death-associated protein kinase Drak that functions as a myosin light chain kinase. We found that Drak biochemically bound to an open mutant of Filamin that resembles the mechanically activated form, partially bound to wild-type Filamin, and did not bind to closed mutant of Filamin. The interaction site was mapped to the intrinsically unfolded C-terminal region of Drak. To study the functional role of Drak-Filamin interaction, we studied two developmental events in which Drak has been earlier shown to be expressed and in which Filamin also functions: early embryonic cellularization and indirect flight muscle development at pupal stages. We found partial colocalization between Drak-GFP and Filamin-mCherry during the initiation of cellularization furrow and at the time of myotube attachment site maturation in tendon cells. However, functionally, we could not show direct correlation between Filamin and Drak. Our studies reveal interesting new expression patterns of Drak during *Drosophila* development and provide detailed information about Filamin localization during IFM development.

KEY WORDS: Mechanosensing, Mechanotransduction, Cytoskeleton, Actin, Filamin, Cellularization, Muscle development, *Drosophila melanogaster*

## INTRODUCTION

Mechanosensing involves proteins that change shape when force is applied (Sala et al., 2024). Mechanical force can alter protein function or expose hidden binding sites, allowing other proteins to interact and initiate a signaling cascade known as mechanotransduction. This process can produce biochemical signals, including post-translational

[1]Department of Biological and Environmental Science, and Nanoscience Center, University of Jyväskylä, Jyväskylä FI-40014, Finland. [2]Department of Physiology, Development and Neuroscience, University of Cambridge, Cambridge CB2 3EG, UK.
*Present address: Bristol Genetics Laboratory, Southmead Hospital, Bristol BS10 5NB, UK.

‡Author for correspondence (riku.o.korkiamaki@jyu.fi)

R.O.K., 0009-0009-3113-8567; C.T., 0000-0002-9554-518X; H.J.G., 0000-0002-3039-3015; J.Y., 0000-0003-4627-021X

modifications (Sala et al., 2024). There are numerous molecules with mechanosensitive properties. There are at least four families of mechanically regulated ion channels that sense membrane pressure changes in response to forces in the cell membrane (Árnadóttir and Chalfie, 2010; Douguet and Honoré, 2019). However, many mechanosensors initiate mechanotransduction processes at cell-cell or cell-extracellular matrix attachment sites or along cytoskeletal protein filaments that are linked to these sites (Vogel and Sheetz, 2006). The cytoskeleton also provides a direct mechanical link between various parts of the cell, for instance between the cell surface and the nucleus (Fedorchak et al., 2014).

Mechanotransduction plays an important role in a wide variety of developmental processes ranging from one-cell embryo development (Duch et al., 2020), embryo compaction at eight- to 16-cell stage (Firmin et al., 2024), to control of tissue shape and size (Vignes et al., 2022). In pathological states, mechanical tissue response such as tissue stiffening and encapsulation of damaged areas are part of defense mechanisms but can also lead to extensive tissue fibrosis and functional failure (Di et al., 2023). Even though many mechanosensing and mechanotransduction pathways are known, we still do not completely understand many of the underlying molecular mechanisms. This is why we, in this paper, study the mechanotransduction partners of the actin cross-linking protein Filamin.

Filamins are dimeric, actin filament crosslinking proteins involved in mechanosensing. Three mammalian Filamins are all essential (Dalkilic et al., 2006; Feng et al., 2006; Hart et al., 2006; Zhou et al., 2007) and have partially distinct, partially overlapping, roles in tissue development (Razinia et al., 2012; Zhou et al., 2010). Human congenital Filamin mutations cause malformations in neurons, bones, and epithelial tissues, as well as in cardiac muscle and skeletal muscle. Many of these diseases are assumed to be caused by alterations in mechanotransduction (Sutherland-Smith, 2011). Filamins are composed of an N-terminal actin-binding domain and up to 24 immunoglobulin-like Filamin repeat domains (Fig. 1A). The mechanosensory region (MSR) at the C-terminal part of Filamin contains two domain pairs that normally mask their protein interaction sites (Fig. 1B, left) (Heikkinen et al., 2009; Lad et al., 2007). Single-molecule experiments (Rognoni et al., 2012) and live-cell fluorescence-based assays (Ehrlicher et al., 2011) have demonstrated that these interaction sites can be unmasked by 3.9 pN forces or by myosin activation. When unmasked (Fig. 1B, right), the mammalian Filamin MSR can bind cell surface adhesion receptors, such as integrins (Kiema et al., 2006) and platelet von Willebrand factor receptor (Nakamura et al., 2006), cytoskeletal regulators FilGAP (Ohta et al., 2006), migfilin (Lad et al., 2008), Fimbacin, Smoothelin (Wang and Nakamura, 2019), mRNA stabilizing proteins La related protein 4 (LARP4) (Mao and Nakamura, 2023) and Ras GTPase-activating protein-binding protein 1 (G3BP1) (Feng et al., 2023), as well as a Hippo-pathway scaffolding protein Salvador homology 1 (SAV1) (Zhang et al., 2023).

*Drosophila* has one Filamin gene [*cheerio* (*cher*)], the protein products of which have a similar domain structure to mammalian

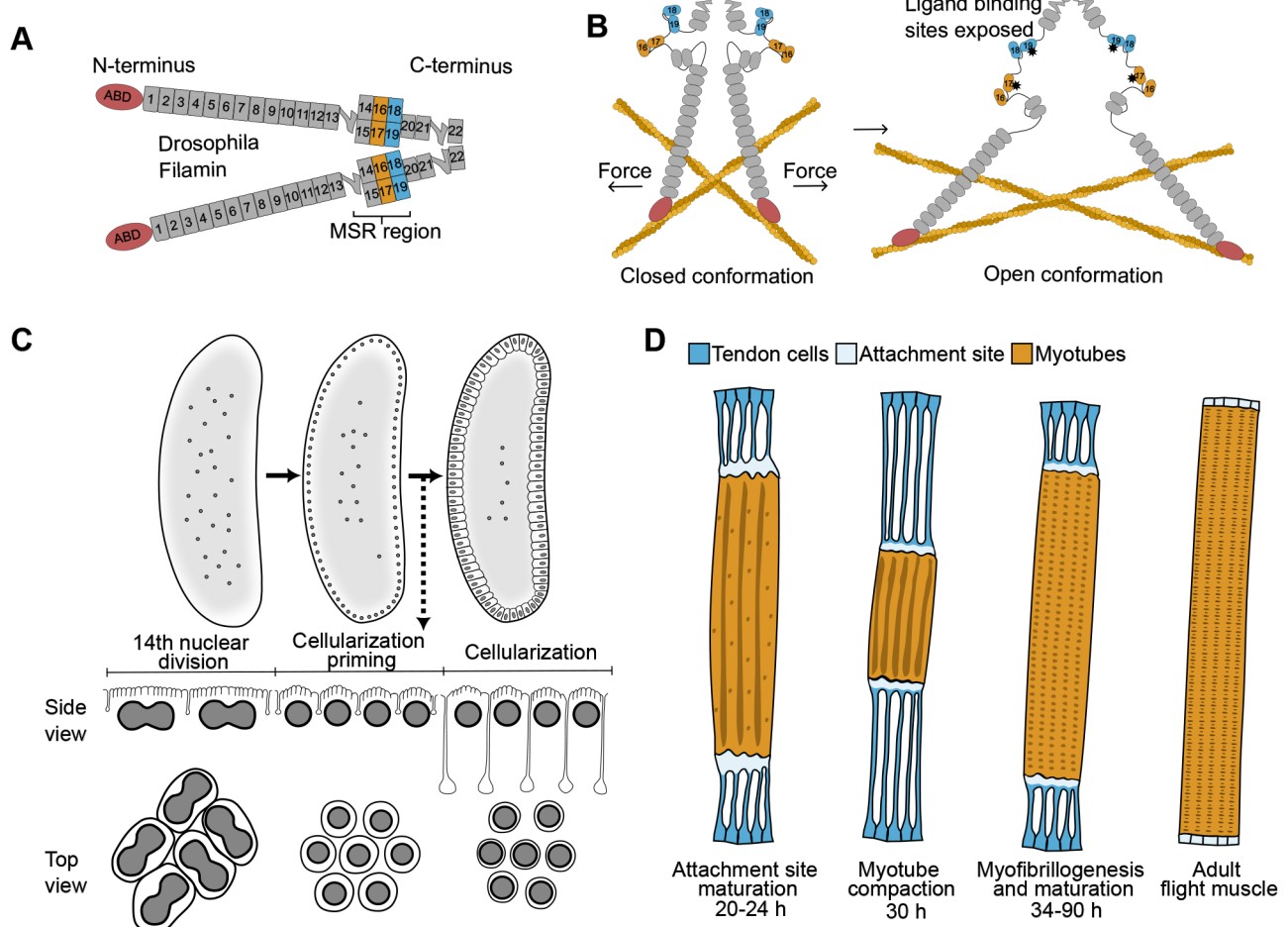

**Fig. 1. Schematics of Filamin protein structure and the *Drosophila* tissue development events studied in this report.** (A) *Drosophila* Filamin (Cheerio) and mammalian Filamins share dimeric structure and domain composition with the main difference that mammalian filamins have 24 filamin repeats and *Drosophila* Filamin has 22. The domain numbering of *Drosophila* Filamin is shown. Actin-binding domains (ABD) are shown in red, prototypical filamin repeats in gray, and mechanically regulated domain pairs at C-terminal mechanosensory region (MSR) in orange and light blue. The two flexible hinge regions of each monomer are indicated. (B) Model of the mechanically opening of the interaction sites at the mechanosensory region. In the closed conformation (left), protein binding sites are masked by protein sequences from the neighboring domains. Actomyosin contraction favors the to open conformation (right) with exposed interaction sites (marked with asterisks). (C) *Drosophila* cellularization takes place after syncytium formation (upper left) and organization of nuclei close the to the plasma membrane (upper middle). After the 14th nuclear division of the embryo (bottom left), initial plasma membrane invaginations are formed between each nucleus, and an actomyosin ring is formed at the tip of the invaginations (bottom middle). The ring moves in between the nuclei with the membrane and eventually contracts to close the membrane under the nuclei (bottom right). Filamin is localized at the actomyosin ring shown in the top view. (D) During pupal indirect flight muscle development, myoblasts fuse to form myotubes that extend and attach to tendon cells linking to the thorax epithelium. At the time of attachment site maturation (20-24 h APF), the contractile forces within the myotube-tendon cell system are low but gradually increase leading to myotube compaction (30 h APF). After the compaction (34-90 h APF), myotube sarcomere maturation and myofibrillogenesis continue until the time of adult enclosure (rightmost image).

Filamin. The other *Drosophila* Filamin like-gene [*jitterbug* (*jbug*)] encodes proteins that appear not to have dimerization domains similar to mammalian Filamin (Mulder et al., 2025). In this paper, we call *cher* gene products *Drosophila* Filamin. The name Cheerio comes from its function and localization in the ring-like actin structure – the ring canal – that is formed between the germ-line derived nurse cells and the oocytes during egg chamber development (Sokol and Cooley, 1999). In addition to this, Filamin is involved in other functions in *Drosophila*, including syncytial embryo cellularization (Krueger et al., 2019), epithelial cell migration (Sokol and Cooley, 2003), motor neuron axon guidance (Zheng et al., 2011), immune cell differentiation (Rus et al., 2006), and muscle development (González-Morales et al., 2017). For instance, during embryonic cellularization, Filamin localizes to the actomyosin ring that drives invagination of the plasma membrane between the

nuclei, and Filamin knockdown leads to cellularization failure in 80% of the embryos (Krueger et al., 2019). During *Drosophila* adult indirect flight muscle (IFM) development, Filamin knockdown causes Z disc defect and flightless flies (González-Morales et al., 2017). In addition to Z disc, Filamin is also localized at the flight muscle attachment site, and a C-terminal deletion of Filamin causes disruption of the attachment (Green et al., 2018). At least in ring canal development (Huelsmann et al., 2016) and flight muscle development (Fisher et al., 2024), mutations in the C-terminal MSR cause observable phenotypic effects.

Myosin contraction is often regulated as part of mechanotransduction pathways. Non-muscle myosin-2 is activated via phosphorylation of its regulatory light chain at residues Thr18 and Ser19 (in human and mouse) (Vicente-Manzanares et al., 2009), which correspond to residues Thr20 and Ser21 in the *Drosophila* ortholog

Spaghetti squash (encoded by *sqh*). Mechanosensing at cell adhesion protein complexes activates Rho kinase (ROCK) that increases myosin regulatory light chain phosphorylation mainly via inactivation of its phosphatase or by Ca2+/Calmodulin-mediated activation of Myosin light chain kinases (MLCK) (Zuidema et al., 2020). In addition to this, other serine/threonine kinases without the Ca2+/Calmodulin regulatory domains can also phosphorylate the myosin light chain. In human and mouse T-cells, Death-associated protein kinase related apoptosis-inducing protein kinase 2 (DRAK2) directly phosphorylates myosin light chain, and DRAK2-deficient T-cells have reduced motility and T-cell receptor microclustering (Wilander et al., 2024).

*Drosophila* Drak, the single ortholog of human DRAK1 and DRAK2, has a role during *Drosophila* cellularization. Cellularization occurs during *Drosophila* early embryo development, when the embryo undergoes 14 rounds of nuclear divisions (Sokac et al., 2023). Nuclear cycles 1-7 occur in the interior of the embryo, after which the nuclei are transported to the periphery and the divisions continue. Cellularization begins at nuclear cycle 14, when the plasma membrane invaginates between each nucleus resulting in a monolayer of approximately 6000 cells (Sokac et al., 2023). A schematic of the cellularization process is shown in Fig. 1C. At the basal tip of the invaginating furrows, an actomyosin ring forms, the contraction of which is tightly regulated throughout the process. Early contraction of the ring is driven by non-muscle myosin (Chougule et al., 2016; Xue and Sokac, 2016), whereas the later closure of the ring depends on actin depolymerization (Xue and Sokac, 2016). Drak null embryos show impaired ring contractility during early cellularization and reduced phosphorylation levels of non-muscle myosin (Chougule et al., 2016).

In addition to early embryos (Chougule et al., 2016) and wide expression in epithelial tissues (Neubueser et al., 2010), Drak mRNA is expressed during specific stages of *Drosophila* IFM development (Spletter et al., 2018). IFM development begins 12 h after pupal formation (APF) with the fusion of myoblasts to form myotubes (Weitkunat and Schnorrer, 2014). The following steps include the splitting of the original IFM myotubes into two and formation of firm integrin-containing attachments to tendon cells (Fig. 1D). The attachment site maturation between 20 and 24 h APF is followed by myotube contraction, which continues until 30 h APF (Weitkunat et al., 2014). This marks the transition of myotubes to immature myofibers when myofibril assembly and sarcomerogenesis is initiated (Lemke and Schnorrer, 2017; Luis and Schnorrer, 2021; Weitkunat and Schnorrer, 2014). Interestingly, the spike in Drak mRNA expression coincides with myotube compaction, occurring between 24 and 30 h APF (Spletter et al., 2018).

In this study, we report a biochemical interaction between *Drosophila* Filamin and Drak. This interaction is facilitated with a mutant that opens Filamin's C-terminal mechanosensitive sites. We tested the hypothesis that Filamin and Drak are mechanotransduction partners during cellularization and during IFM development.

## RESULTS
### Drak binds to open Filamin

Filamins are known mechanosensors that change conformation in response to piconewton magnitude pulling force (Rognoni et al., 2012). Their MSR is typically masked, making it challenging to identify binding partners using conventional screening methods. To overcome this, we used bacterially expressed Filamin fragments with altered MSRs that expose two masked binding sites. To identify mechanically regulated binding partners of Filamin, we ordered a yeast two-hybrid screen with the *Drosophila* Filamin MSR. As bait, we used *Drosophila* Filamin 16-19 domains with

previously characterized open mechanosensor mutation exposing binding sites in domains 17 and 19. The screen resulted in 249 sequenced clones that mapped to 58 protein-coding genes in the sense orientation. While most of the hits were single fragments, several overlapping clones were identified, corresponding to three genes: Death-associated protein kinase related *Drak*, the BAR-like FCH domain-containing protein *NOSTRIN* and the cohesin-complex subunit *vtd* (Table S1). Since Drak has been previously linked to cytoskeleton function (Chougule et al., 2016), we selected Drak for further studies. The eight overlapping hits of Drak mapped to the COOH-terminal region of Drak and the minimal common sequence contained amino acid (aa) 360-476 (Fig. 2A).

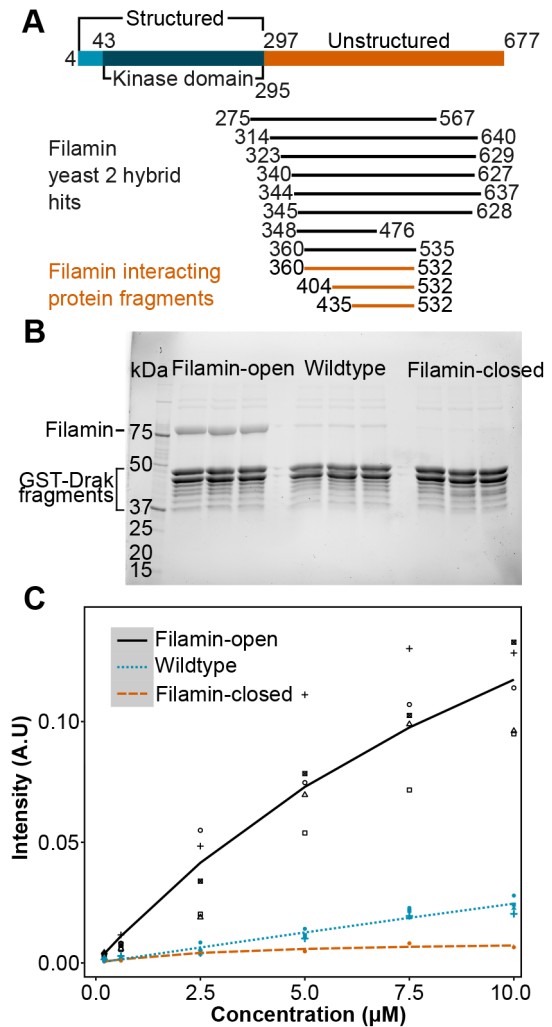

**Fig. 2. Drak binds to open Filamin.** (A) Schematic diagram of Drak kinase as depicted by Alphafold structure (Identifier AF-Q0KHT7-F1), highlighting structured (blue) and unstructured (orange) regions, the kinase domain (dark blue), and their amino acid positions. Drak fragments interacting with Filamin are shown in black (yeast two-hybrid data) and orange (protein binding assay data). (B) SDS-PAGE analysis of a pull-down experiment showing binding of Drak to Filamin-open (lanes 2-4), Filamin-wild type (lanes 6-8), and Filamin-closed (lanes 11-12). The positions of five Filamin fragments and GST-Drak fragments are shown on the left. Three technical replicates of the pull-down experiment with GST-Drak (435-532) and Filamin-open, wild type and closed are shown. (C) Quantation of Filamin binding with different concentrations. A.U., arbitrary units. All measurement points are mean of triplicate measurements. Number of biological replicates in each group: Filamin-open, 5; wild type, 4; Filamin-closed, 1. All SDS-PAGE gels used for the analysis are shown in Fig. S1.

To biochemically verify the yeast two-hybrid interaction, we attached short GST-Drak fragments on glutathione Sepharose and used these in a pull-down system to test the interaction with Filamin. In this assay, we used the previously characterized five-domain fragment of *Drosophila* Filamin (Huelsmann et al., 2016), because both the five-domain wild-type fragment and the open and closed mechanosensory mutants could be readily expressed and were soluble. The five-domain Filamin fragment was also readily distinguished in SDS-PAGE from the GST-Drak and its degradation products. The minimum Drak fragment that could be expressed and interacted with the Filamin-open protein contained aa 435-532 (Fig. 2A). At 10 µM concentration, Filamin-wild type showed weaker interaction with this Drak fragment than Filamin-open, whereas Filamin-closed did not interact (Fig. 2B). This difference could be reproduced with concentrations between 0.2 µM and 10 µM (Fig. 2C), but accurate Kd (dissociation constant) could not be determined because of experimental variation.

### Drak and Filamin in early embryo development

As Drak has a known function during cellularization, we next tested if it has similar localization with Filamin or its substrate myosin regulatory light chain, Spaghetti squash, in early *Drosophila* embryos. To do this, we genomically inserted Green fluorescent protein (GFP) in frame to the COOH-terminus of Drak and followed Drak-GFP in live time-lapse confocal imaging simultaneously either with Filamin-mCherry or Spaghetti squash-mCherry. At the onset of cellularization, all three proteins were recruited to the cortical region of the embryo (Fig. 3A, XZ view). Drak recruitment followed Filamin and Spaghetti squash with a slight delay and remained high at the cortex for longer than the two others (Fig. 3B). This recruitment was linked to the last (14th) nuclear division. At the beginning of the division, Filamin and Spaghetti squash weakly marked the transient metaphase furrows at the plasma membrane between nuclei, but Drak was not detected in these structures (Fig. 3A, 0 min and 3 min). However, after the division, Drak was seen in the forming cellularization furrows together with Filamin and Spaghetti squash (Fig. 3A, 6 min and 9 min, marked with white arrowhead in Drak XY view). When the furrows proceeded, Drak remained at the cortical region (Fig. 3A, black arrow on the right, XZ view, 21 min) whereas Spaghetti squash strongly localized to the basal region of the furrow (Fig. 3A, black arrowhead on the right, XZ view, 21 min), and Filamin was seen in both cortical and basal areas. To confirm the specificity of Drak-GFP and Filamin-mCherry signals, we imaged single-label control embryos lacking either transgene (no Drak-GFP or no Filamin-mCherry) using identical imaging parameters and brightness scaling (Fig. 3C). The surface of the embryo showed autofluorescence on both channels, but we observed no significant autofluorescence elsewhere on either channel.

Later during cellularization, as the basal tip of the cellularization furrow containing the actomyosin ring advanced between the nuclei, Drak-GFP remained mainly diffuse in the cytoplasm. However, a small fraction of Drak-GFP was detected in the actomyosin ring together with Filamin (Fig. 3D). This partial localization of Drak was weakly seen in the confocal z-projections (Fig. 3D, middle row, arrowheads on the left), and in the intensity profiles (Fig. 3D, 8.5 µm and 15.5 µm). When the actomyosin ring contracted below the nuclei, Drak fluorescence was not associated with the ring (Fig. 3D, 21 µm).

As the images from time-lapse confocal recording had too low signal intensity for rigorous pixel-to-pixel colocalization analysis, we used higher laser power to take individual confocal images of the embryos right after the 14th nuclear division, corresponding to the 6 min time point in Fig. 3A. We combined 37 images from nine individuals for colocalization analysis and found that Drak-GFP and Filamin-mCherry showed high pixel-to-pixel overlap [Mander's coefficients; M1 mean was 0.998 with a 95% confidence interval (CI) of 0.997-0.999 ($P$=2.2e-16, one-sample $t$-test), M2 was 0.970 [0.946, 0.995] ($P$=2.26e-13)] and a small-to-moderate intensity correlation with Pearson's correlation coefficient mean of 0.325 [0.243, 0.402] ($P$=2.33e-5). Fig. 4 shows a representative image pair with the corresponding intensity dot plot. Representative images from all samples and the control samples with individual fluorophores that were used for thresholding the intensities are shown in Fig. S3. Correlation coefficient values from all images are given in Table S2. Colocalization analysis was validated by randomizing pixels in the Drak-GFP channel, resulting in close to zero thresholded Pearson correlation coefficient (tPCC) values (Table S2).

To test if reducing Filamin-Drak interaction would cause similar changes in cellularization dynamics as Drak depletion, we compared the actomyosin ring circularity in flies expressing Filamin-closed MSR mutation to flies expressing wild-type Filamin-GFP or Filamin-GFP with Drak-knockout (KO). We were able to reproduce the previously reported (Chougule et al., 2016) non-circular actin ring phenotype with Drak-KO, but Filamin-closed MSR actin rings did not differ from Filamin-wild type rings (Fig. 5A,B). In line with this, myosin light chain phosphorylation was not reduced in Filamin-closed embryos, even though a reduction was detected in Drak-KO embryos (Fig. 5C,D).

### Drak and Filamin in *Drosophila* flight muscle development

We next studied the potential Filamin-Drak interaction in *Drosophila* developing muscles as Drak mRNA is detected during IFM development (Spletter et al., 2018). Living pupae were imaged by taking a confocal time series of the fly thorax.

We used Spaghetti squash-mCherry to localize the developing myotubes at the time of the myotube compaction, 33 h APF (Fig. 6A, left). Filamin-mCherry, on the other hand, marked the tendon cells (Fig. 6A, right). At this time point, Drak was mainly detected in the myotubes (Fig. 6A,B) and weakly in the tendon cells (arrowhead in Fig. 6B). When we followed Drak-GFP, we found that its intensity increased in myotubes during the compaction stage (Fig. 6B,D; Movie 1). Highest Drak-GFP intensity was observed at 35 h APF, coinciding with the maximum compaction of the developing myotube between 33 and 35 h APF (Fig. 6D). A weak Drak-GFP signal remained in the tendon cells through these stages (Fig. 6B, 33 h, arrowhead). During the compaction stage, Drak-GFP distribution was diffuse, and no structural concentration to attachment sites or filamentous structures was observed (Fig. 6B). Filamin, however, was detected in the tendon cells in filamentous structures (Fig. 6C, 33-37 h, arrowhead) and appeared in the developing myotubes with a patterned distribution immediately after maximal compaction of myotubes, apparently coinciding with sarcomerogenesis (Fig. 6C, 35-37 h, star).

Before myotube compaction, between 21 and 23 h APF, when the muscle-tendon attachment sites undergo maturation, a transient Drak-GFP signal was also detected in tendon cells (Fig. 7A-C). This signal appeared at about 21 h APF (Fig. 7A, left), and, at 23 h APF, Drak-GFP was localized at the tendon cell attachment site together with Filamin-mCherry (Fig. 7A, right, arrows). To confirm that this Drak-GFP signal was specific, we imaged pupae without the Drak-GFP transgene, expressing the Filamin-mCherry allele only (Fig. 7B). Using identical imaging parameters and brightness scaling, we observed significant autofluorescence in the GFP channel, mainly in free-moving cells (Fig. 7B, 21-23 h, no Drak-GFP). These cells were occasionally found over the tendon region,

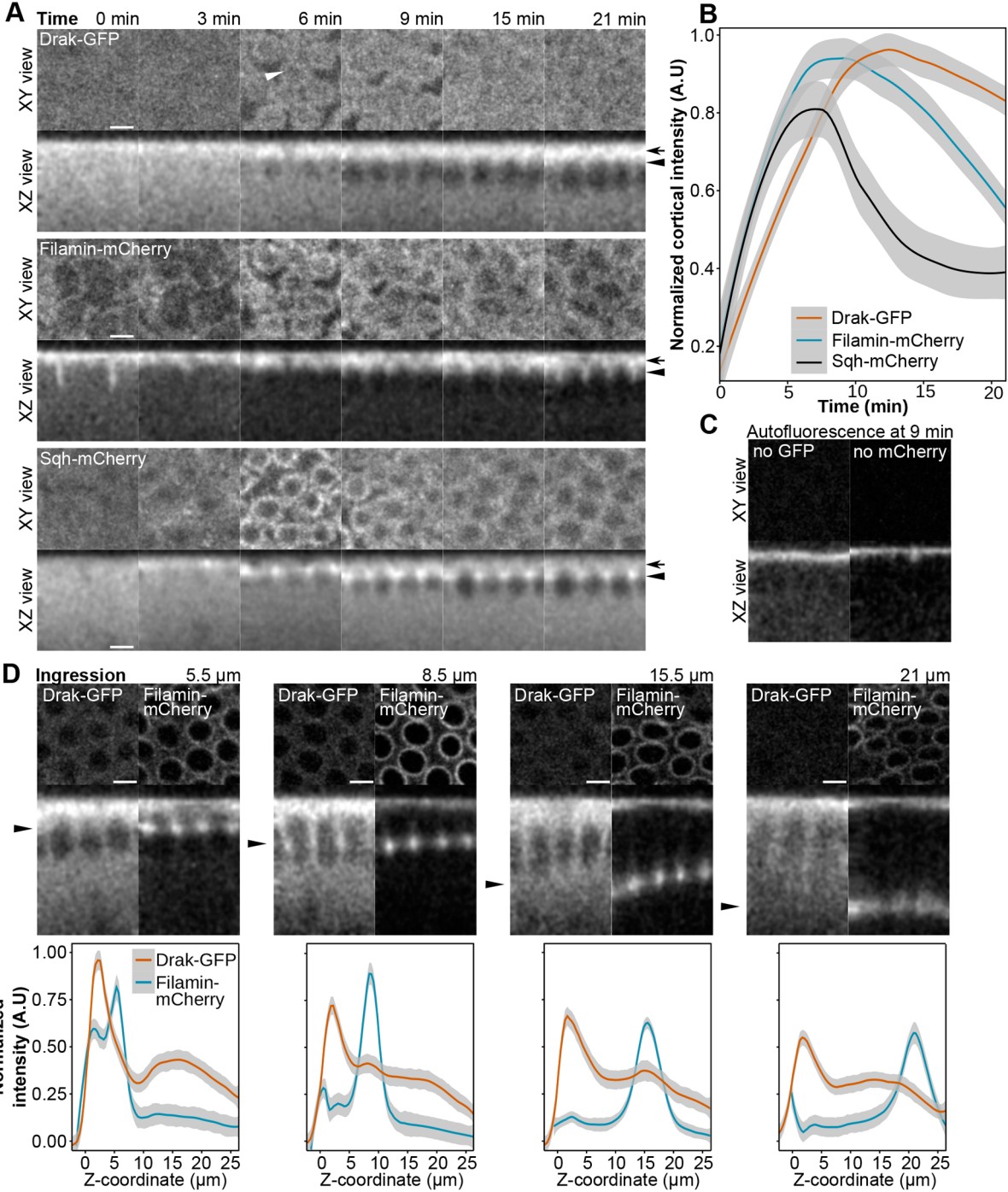

**Fig. 3. Drak, Filamin and Spaghetti squash (Sqh) localization during cellularization.** (A) Distribution of Drak-GFP (top panel), Filamin-mCherry (middle panel), and Sqh-mCherry (bottom panel) during the first 0-21 min starting from the last nuclear division. Confocal microscope fluorescence signals are shown in top down (XY view; five confocal planes) and sagittal sections (XZ view; projection of 7.5 µm thickness). Drak-GFP and Filamin-mCherry images are from the same sample. The arrows on the right point to the cortical region and the arrowheads to the basal region of the cellularization furrows. (B) Normalized and background-corrected cortical intensity of Drak-GFP (n=6), Filamin-mCherry (n=3), and Sqh-mCherry (n=3) during early stages of cellularization. (C) Single-labeled control embryos lacking either Drak-GFP (no Drak-GFP) or Filamin-mCherry (no Filamin-mCherry), showing autofluorescence in the GFP and mCherry channels at the 9 min time point. Imaging parameters and brightness scaling were identical between panels A and C for the corresponding channels. (D) Top-down and sagittal sections of Drak-GFP and Filamin-mCherry from later stages of cellularization (top panel), during actomyosin ring ingression of 5.5 µm, 8.5 µm, 15.5 µm and 21 µm from the surface of embryos. Arrowheads in the sagittal sections point to the location of the actomyosin ring. The bottom panel shows the normalized and background-corrected intensity profiles of Filamin-mCherry and Drak-GFP from timelapse recordings (n=3) that were aligned using the Filamin-mCherry signal peak at the actomyosin ring position. Scale bars: 4 µm.

but we were able to distinguish them from the tendon-associated signal because the signal was typically brighter and round, whereas the tendon-associated signal was more uniform and followed the

Filamin-mCherry-marked attachment site structure. Consistent with this, Drak-GFP signal appeared at the expected location along Filamin-mCherry with similar intensity and morphology across eight pupae

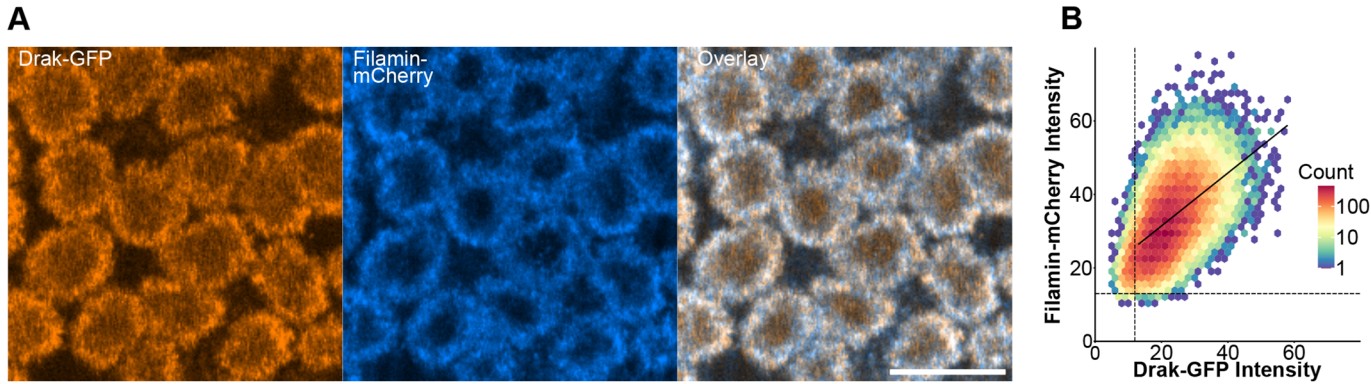

**Fig. 4. Colocalization analysis during the priming phase of cellularization show strong overlap between Drak and Filamin.** One representative image (sample 8, image 1, see Fig. S3) is shown. (A) Drak-GFP (orange) and Filamin-mCherry (blue) show distinct distribution around the newly formed nuclei. Overlapping signal is white (overlay). Scale bar: 10 µm. (B) Density dot plot of Drak-GFP and Filamin-mCherry pixel intensities. Dashed vertical line shows threshold for Drak-GFP channel, and dashed horizontal line shows threshold for Filamin-mCherry. The linear regression line is plotted using above-threshold pixel values.

(three complete-thorax time series and five high-resolution single-plane datasets used for colocalization). This is especially apparent when comparing the attachment sites between samples with and without Drak-GFP at the 23 h time point (Fig. 7A,B, arrows). We similarly tested the specificity of Filamin-mCherry. Filamin-mCherry-negative controls showed only a few bright autofluorescent structures in the mCherry channel that did not interfere with tendon or myotube imaging (Fig. 7C, no Filamin-mCherry).

To study the localization of Drak-GFP and Filamin-mCherry more closely in the tendon cell attachment sites at the 23-24 h time point, we acquired high-resolution images with a 63× objective (Fig. 7D). These images revealed a pattern in Drak-GFP distribution, which partially colocalized with Filamin-mCherry. To quantify the colocalization, we performed a pixel-to-pixel colocalization analysis from individual confocal slices combining a total of 17 images from five pupae. The analysis showed high overlap between the fluorophores with weighted Mander's coefficients [M1 mean was 1.00 with a 95% CI of 0.999-1.00 (*P*=4.97e-16, one-sample *t*-test), M2 was 0.870 [0.656, 1.08] (*P*=0.00035), and moderate intensity correlation with weighted and thresholded Pearson's correlation coefficient mean of 0.450 [0.280, 0.592] (*P*=0.00241)] (Fig. 7D; Fig. S4 and Table S3).

To test if the diffuse localization of Drak-GFP and only partial colocalization with Filamin-mCherry could be caused by proteolysis of Drak-GFP protein, we immunoblotted either adult fly extracts or pupal proteins from 28-30 h pupae with anti-GFP. As a control, we used a 45 kDa bacterially expressed GFP fusion protein. We were not able to specifically detect Drak-GFP or GFP degradation product from the Drak-GFP-expressing flies by immunoblotting, apparently due to a low expression level (Fig. S5).

To study if *Drak* and *Filamin* participate in the same processes during IFM development, we examined Drak-KO, Filamin-closed MSR, and a double mutant combining both mutations. We measured the length of the actin-rich muscle attachment sites in adult flies, as earlier studies have shown increased muscle attachment site length in flies with defective Filamin MSR (Green et al., 2018). Filamin-closed MSR mutant showed significantly increased attachment site length, while Drak-KO mutant had a slightly shorter length (Fig. 8A,B). Interestingly, the double mutant showed significantly longer attachment sites compared to the additive effects of Drak-KO and Filamin-closed genotypes, showing an epistatic interaction.

## DISCUSSION

In this study, we report a new interaction between *Drosophila* Filamin and Drak. We show that biochemically Drak preferentially bound to the open MSR of Filamin and less to the wild-type Filamin MSR fragment, and did not bind to closed MSR of Filamin. By inserting GFP in frame to the *Drak* locus, we demonstrate that Drak and Filamin have similar localization at the priming phase of cellularization and in tendon cells at the time of IFM attachment site maturation in pupae. We also found that Drak is expressed in the developing myotube at the time of maximal compaction. When combined in the same flies, Drak-KO and Filamin-closed MSR alleles change the length of the actin-rich layer at the muscle attachment site. Below, we discuss the interaction between Drak and Filamin and Drak-GFP localization studies in early embryo cellularization and in IFM development.

### Filamin-Drak interaction

By combining yeast two-hybrid and biochemical pull-down experiments, we were able to map the sequence required for Filamin binding to a 42 aa (435-476) stretch within the 380 aa-long C-terminal, unstructured part of Drak. This was the overlap between the shortest interacting fragments in yeast two-hybrid and pull-down experiments. Apparently, the 42 aa are not sufficient for binding or sufficiently stable for interaction, since the minimal interacting fragment in yeast two-hybrid screen was 129 aa and in pull-down experiment 97 aa. While the wild-type Filamin fragment binding was just detectable over the background at 10 µM protein concentration, the open-MSR mutant bound about ten times more, and the closed-MSR did not bind (Fig. 2B,C). This suggests that the interaction of Drak and Filamin is specific and mechanically regulated.

Drak can be grouped to the family of death associated protein kinase (DAPK). Typically, this family of proteins contain autoregulatory motifs C-terminal to the kinase domain (Temmerman et al., 2013). Among mammalian death associated protein kinases, DRAK1 and DRAK2 are the only members of the family for which the C-terminal parts have no known motifs (Temmerman et al., 2013). In DRAK1 and DRAK2, the 80-90 aa-long C-terminal parts are predicted to be intrinsically disordered (Uniprot Q9UEE5, human DRAK1; O94658 human DRAK2). In the genus *Drosophila*, this disordered C-terminal part is even longer – 380 aa. According to AlphaFold prediction (identifier AF-Q0KHT7-F1) and by protein disorder prediction programs such FoldIndex and IUPred, this area shares features of

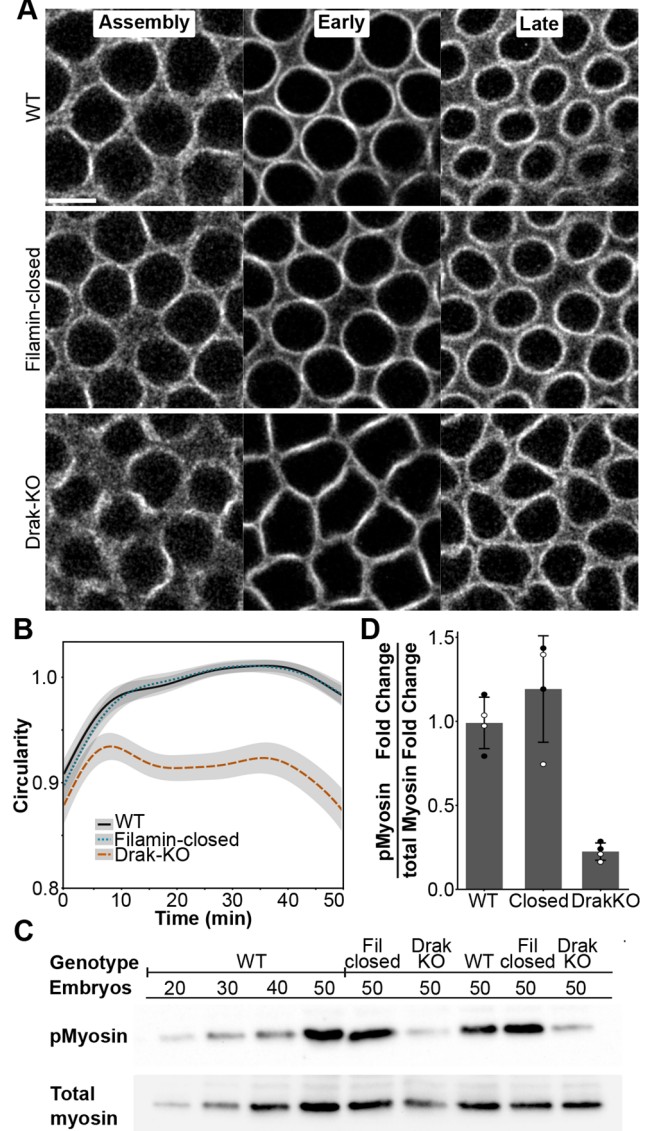

**Fig. 5. Drak knockout causes impaired cellularization dynamics and reduced myosin phosphorylation.** (A) Representative confocal images of actomyosin rings during time points 6 min (left, assembly), 21-27 min (middle, early), and 39-45 min (right, late) in Filamin-wild type (WT), Filamin-closed, and Drak-KO flies, visualized with Filamin-GFP. Note that in Drak-KO embryos the actomyosin rings remain hexagonal, while in the other samples they became rounded already at the early cellularization. Scale bar: 5 μm. (B) Circularity change of actomyosin rings over time in Filamin-wild type (*n*=17), Filamin-closed (*n*=15), and Drak knockout (Drak-KO) (*n*=13) flies. (C) Western blot of phosphorylated Sqh (pMyosin) and total myosin in genotypes Filamin-wild type, Filamin-closed (Fil closed), and Drak-KO. The number of embryos is shown. (D) Quantification of pMyosin to total myosin fold change ratio relative to wild type. Black and white dots represent technical replicates of two different experiments. All gels of replicate analyses are shown in Fig. S2.

intrinsically unfolded protein sequences, because it contains an exceptionally high proportion of polar and charged residues (Uniprot Q0KHT7, *Drosophila* Drak).

### Drak in cellularization

Our results suggest that Drak has a role at the priming phase of cellularization. This is supported by three observations: (1) Drak-GFP was recruited to the cortical region after the last nuclear

division before cellularization; (2) Drak-GFP transiently marked the developing of cellularization furrows at this stage and colocalized with Filamin; and (3) Drak-KO allele causes a defect in the cellularization ring contraction, which starts at the assembly state of the cellularization ring (Chougule et al., 2016; Fig. 3).

Filamin, myosin-2, and Drak are all known components of the actomyosin regulation protein network during cellularization and all three are translated from maternal mRNA (Sokac et al., 2023). However, they are just a part of the large protein repertoire involved in cellularization, which requires spatial and dynamic coordination of various mechanisms such as membrane sculpting, endocytosis, vesicle transport and fusion, microtubules and actomyosin filaments (Sokac et al., 2023). Similar delay of the cellularization contraction as with the Drak mutant, has also been observed with inactive Sqh$^{AA}$ mutation, where the phosphorylated residues are mutated to Ala (Xue and Sokac, 2016). All this is consistent with Drak being a myosin light chain kinase activating myosin contraction at the priming phase and at the early phase of cellularization (Chougule et al., 2016). Interaction of Drak with open Filamin MSR mutation and transient colocalization at priming phase suggest that Filamin might be involved in Drak function at the forming cellularization furrow. Unfortunately, we could not confirm the role of Filamin in Drak function by using the Filamin-closed MSR mutation *in vivo*. We could not observe a consistent change in the actin ring dynamics in embryos with this Filamin mutation. We were able to reproduce the reduced myosin light chain phosphorylation with Drak knockout mutation (Chougule et al., 2016), but we did not detect similar change in phosphorylation with the Filamin-closed MSR mutant. It is possible that our Filamin-closed MSR mutant is still partially functional *in vivo*. This mutant contains masking sequences from the platelet glycoprotein Ib (GPIb) that bind stronger to the mechanosensory sites than the native sequences (Huelsmann et al., 2016). In single-molecule experiments, the displacement of GPIb peptide from human Filamin A domain 21 requires about 10 pN force, whereas the native sequence of domain 20 is displaced at an average of 3.9 pN (Rognoni et al., 2012). We are not aware of direct force measurement performed during *Drosophila* cellularization, but with talin molecule at the muscle attachment sites, forces exceeding 10 pN have been observed (Lemke et al., 2019). Thus, it is possible that the Filamin-closed MSR variant can be at least partly opened in cells and may thus be partially functional. On the other hand, it is possible that during cellularization the possible Filamin-Drak interaction only has a minor role in the regulation of Drak activity at the early phase of actomyosin organization.

One possible strategy to study the role of Drak-Filamin interaction *in vivo* would be to use stronger Filamin MSR mutants. Unfortunately, stronger mutants, such as Filamin ΔMSR lacking six Ig-like domain at the MSR site or Filamin ΔC, lacking the entire C-terminus, could not be used for the live time-lapse recording or timed embryo collection, because they caused significant delay and irregular cellularization, even though a small portion of embryos survive.

### Drak and Filamin in flight muscle development

To study the possible role of Drak and Drak-Filamin interaction in IFM development, we studied the Drak-GFP expression pattern during pupal developmental stages and combined Drak and Filamin mutant alleles. Drak-GFP protein expression was observed in the tendon cells at the time of tendon-myotube attachment site maturation and then peaks at developing IFM myotubes at the stage of maximal myotube compaction. Filamin, on the other hand, could be used as a marker of tendon cells during the tendon-myotube attachment and

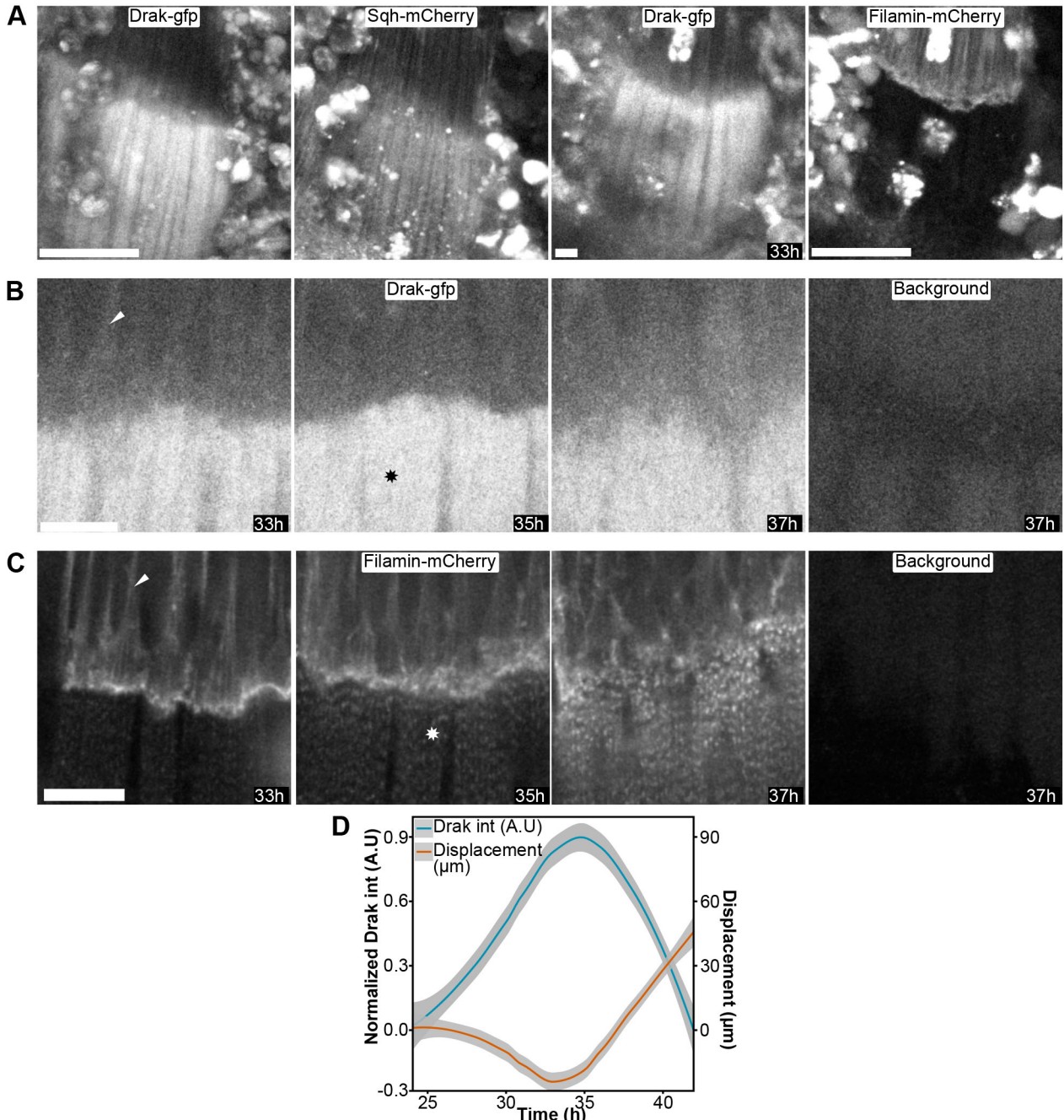

**Fig. 6. Localization of Drak, Filamin, and Spaghetti squash (Sqh) in myotube and tendon cells during indirect flight muscle myotube compaction.**
(A) Distribution of Drak-GFP (first and third column), Sqh-mCherry (second column), and Filamin-mCherry (fourth column) during myotube compaction peak at 33 h after pupal formation (APF) imaged with 20× objective. Two leftmost images (Drak-gfp and Sqh-mCherry) and two rightmost (Drak-gfp and Filamin-mCherry) are from the same fly. Scalebar is 50 µm. (B) Drak-GFP, imaged with 63× objective, and (C) Filamin-mCherry, imaged with 60× objective, in time points 33-37 h. Arrowheads indicate location of tendon cells; stars indicate myotubes. The rightmost panel image in B and C show the background of the corresponding channel. Scalebars in B and C are 10 µm. (D) Normalized Drak-GFP intensity in myotubes (blue, y-axis on the left) and displacement of the developing myotubes (orange, y-axis on the right) plotted against average time APF. The data represent mean±standard error, n=3.

attachment site maturation. Only later, after the maximal compaction of IFM myotubes, Filamin-mCherry fluorescence was detected in the myotubes. Whereas Filamin was apparently associated with the actin cytoskeleton both in tendon cells and later during sarcomerogenesis in the myotubes, the distribution of Drak-GFP remained rather diffused in these tissues apart from the short period of tendon cell attachment site maturation. The attachment site initiation includes formation of filopodial structures by the tendon cells that form integrin-containing contacts with the myotubes (Weitkunat et al., 2014). Apparently, the cells attach to each other via extracellular matrix, but the cellular cytoskeletal structures associated with integrins, and attachment sites may be different in myotubes and tendon cells. It is interesting that Filamin and Drak appear to be components of the attachment site on the tendon side, but not in the myotube. Later, in mature muscle Filamin is a component of the actin-rich layer between the integrin-associated juxta membrane complex and the first Z-line-resembling structure (Green et al., 2018), but we found here that we could not detect Drak-GFP IFM during maturation or in adult muscle. Overall

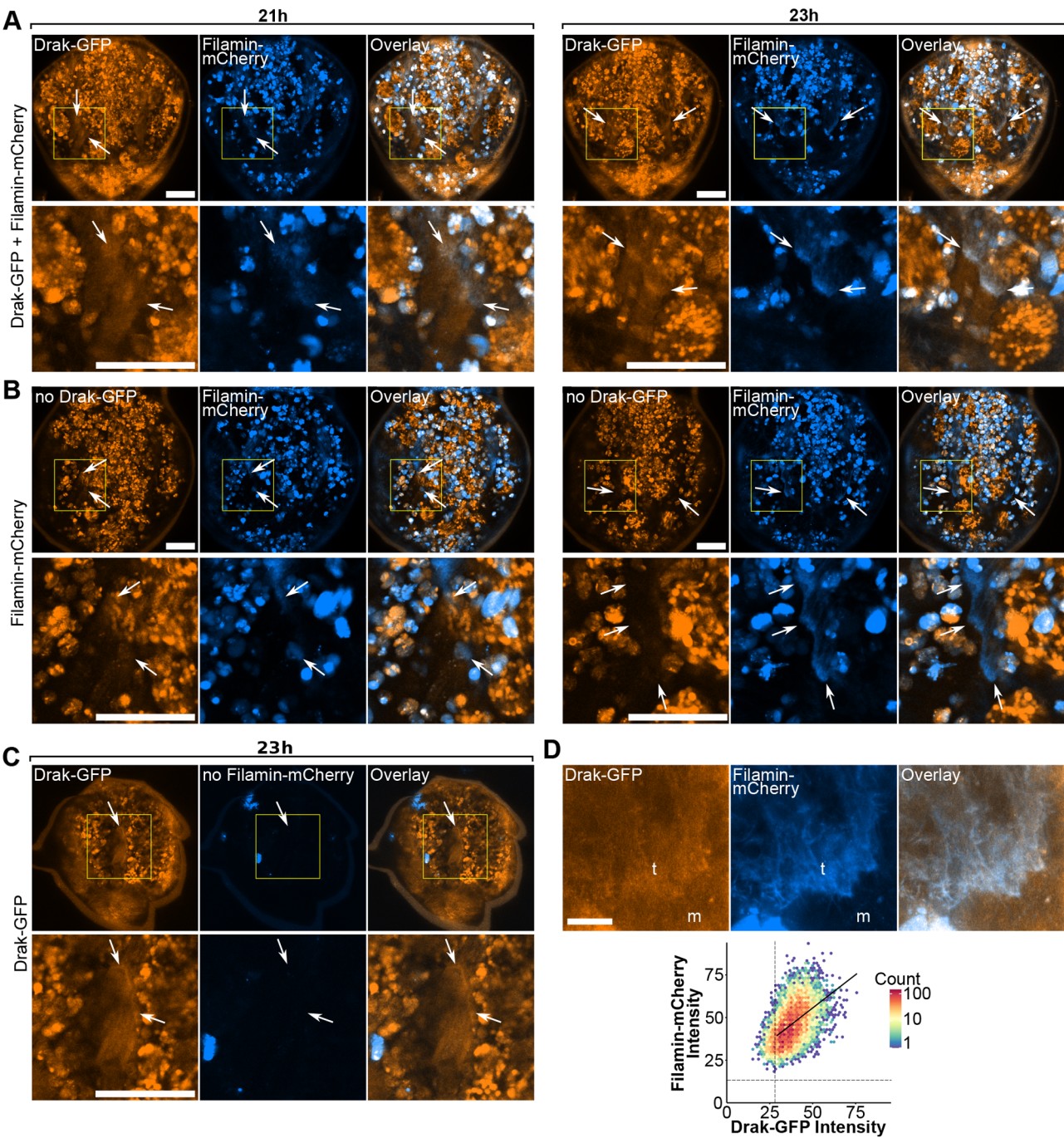

**Fig. 7. Drak and Filamin colocalize at tendons during tendon-myotube attachment site maturation in developing indirect flight muscles.** Overview of *Drosophila* thorax during attachment site maturation. Attachment sites become visible between 21 h (left panels in A) and 23 h (right panels in A and B). Tendon cells and myotubes are visualized by Drak-GFP (orange) and Filamin-mCherry (blue). Overlapping signal is white (overlay). Imaging parameters and brightness scaling in A-C are identical. (A) Fly thorax with both Drak-GFP and Filamin-mCherry alleles. Upper panel shows the overview of the thorax. Boxed area, enlarged in lower panels, shows the location of attachment site, indicated by arrows. (B) Fly thorax with only Filamin-mCherry allele. Autofluorescence of the GFP channel (no Drak-GFP) shown in orange. (C) Fly thorax with only Drak-GFP allele. Autofluorescence of the mCherry channel (no Filamin-mCherry) shown in blue. (D) Representative images used for quantitative colocalization analysis of single confocal slices during attachment site maturation. Tendons (t) and myotubes (m) are labeled. Lower image shows the density dot plot of Drak-GFP and Filamin-mCherry pixel intensities. Dashed vertical line shows threshold for Drak-GFP channel, and dashed horizontal line shows threshold for Filamin-mCherry. A linear regression line was plotted using only pixels that were above the threshold in both channels. Representative images of each sample are shown in Fig. S4. The images in D are from sample 3, image 1. Scale bars: 100 µm in A-C; 10 µm in D.

Drak-GFP levels remain very low both during pupal muscle development and during cellularization, and we could not detect Drak-GFP by immunoblotting with anti-GFP antibody. This raises a question about the turnover rate of Drak protein. It is possible that because of the long intrinsically unfolded C-terminal region consisting about half of the protein length, Drak protein is rapidly

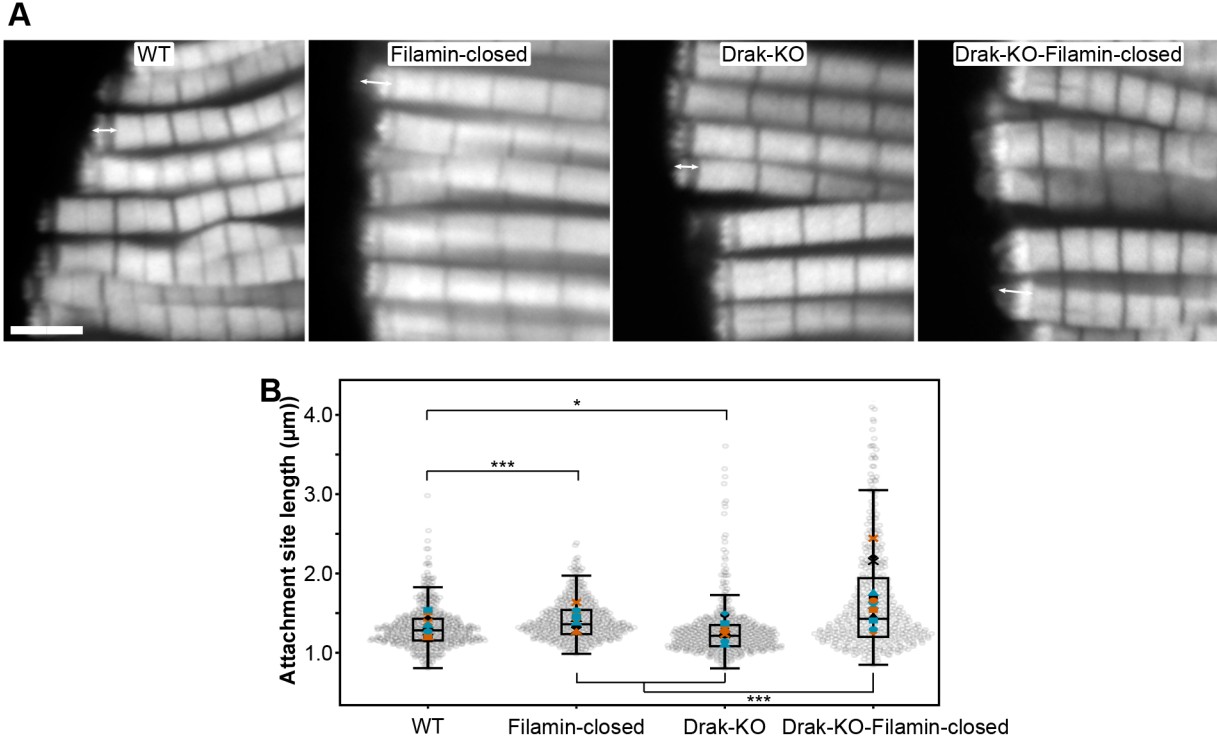

**Fig. 8. Attachment site length in adult flies.** (A) Representative confocal images of dissected adult IFMs. Double arrow shows the location of muscle attachment sites. Scale bar: 5 µm. (B) Distribution of attachment site lengths in Filamin-wild type (*n*=13), Filamin-closed (*n*=13), Drak-KO (*n*=14), and Drak-KO-Filamin-Closed (*n*=15) flies. Approximately 20 attachment sites were measured per fly. Gray dots represent individual attachment site measurements, while colored symbols indicate the mean values for each fly. The boxplot's central horizontal line represents the median for each genotype, the box edges mark the lower and upper quartiles, and the whiskers extend to the minimum and maximum data points, excluding outliers. *P*<0.05, ***P*<0.001. *P*-values were computed using two-sided Wald z-tests for the specified contrasts from the fitted gamma generalized linear mixed-model (log link), using the model-based standard errors and adjusted with Bonferroni correction.

degraded and thus short-lived, explaining its diffused appearance and low sensitivity to immunodetection. In general, proteins containing large disordered regions have shorter half-lives than more stably structured proteins (Fishbain et al., 2015; van der Lee et al., 2014), although the correlation can change drastically in different contexts, such as different tissues or cell cycle (Hasper et al., 2023). In human cells, DRAK1 has been reported to undergo CUL3-SPOP-dependent K48 polyubiquitination, resulting in proteasomal degradation (Pang et al., 2022) Given that the CUL3-SPOP system is conserved in *Drosophila* (Roadkill/HIB) E3 ligase (Pan et al., 2017), it's possible that *Drosophila* Drak is also targeted through related ubiquitin-ligase machinery.

In our experiments, the maximal myotube contraction occurred between 33 and 35 h APF at 24°C. This corresponds to the 30 h APF point reported earlier at 27°C (Weitkunat et al., 2014). This compaction was associated with increased Drak-GFP signal. Earlier, similar increase of Drak mRNA has been observed with transcriptomics sequencing (Spletter et al., 2018). Our analysis confirms the transcriptomics results at protein level, provides better temporal resolution of protein expression changes and shows the expression also in tendon cells. This suggests that Drak has some role in myotube and tendon cell compaction.

Earlier, we reported defects in adult IFM caused by Filamin ΔC mutation and Filamin RNA interference (Green et al., 2018). By extending the analysis with more samples, we now report that Filamin-closed MSR also causes a lengthening of the actin-rich layer of the IFM attachment site. Drak-KO alone caused a minor decrease in the attachment site. However, the combined effect Filamin-closed MSR and Drak-KO mutation was statistically larger than the sum of the individual effects. This suggests that Drak and the Filamin MSR have a role in the same process. Colocalization of Drak and Filamin in the tendon cells at the time of myotube attachment suggests that this might be an event, when the two proteins have synergistic function.

Mapping the Filamin interaction to the C-terminal part of Drak raises the possibility that Filamin binding could regulate Drak kinase activity analogously to how Ca2+/Calmodulin regulates the kinase activity of DAPKs and myosin light chain kinases containing an autoregulatory domain C-terminal to the kinase domain (Temmerman et al., 2013). However, at this point, this kind of regulation pathway is only speculative. We have no experimental data that Drak kinase activity could be post-translationally regulated. Some phosphorylation sites have been observed in Drak (Zhai et al., 2008) and in human DRAK1 and DRAK2 (Ochoa et al., 2020; summarized in UniProt), but we do not have a mechanistic explanation how these sites at the N-terminal or C-terminal parts of proteins could regulate the enzymatic activity. As discussed above, we could not demonstrate that filamin mutations specifically change non-muscle myosin phosphorylation during cellularization, even though Drak is involved in this process. This, together with the colocalization patterns that are evident in only specific developmental sub-processes, the largely distinct phenotypes of single mutants but synergistic phenotypes observed in certain contexts, suggests that Filamin is likely not a major regulator of Drak in non-muscle myosin activation. Instead, the function of the possible interaction is likely context dependent and operates in short developmental time frames. One possibility is that Filamin recruits or scaffolds Drak at specific subcellular sites, enabling localized signaling that is important in particular processes.

In conclusion, we found *in vitro* biochemical interaction between Drak and the Filamin MSR. Mutations that open the mechanosensory sites, enhanced the interaction and mechanosensory site-closing mutations diminished the interaction. Our studies with Drak-GFP shed new light on Drak expression and localization. Notably, Drak localization pattern transiently followed Filamin and myosin in the apical region of the forming cellularization furrows during the early stages of cellularization. Furthermore, Drak and Filamin colocalize in tendon cells at the time of IFM myotube-tendon cell attachment Even though Drak expression level is low in the tissues studied here, we believe that the Drak-GFP line generated here is a useful tool for further analyses.

## MATERIALS AND METHODS

### Expression and purification of the recombinant filamin and Drak fragments

The five-domain *Drosophila* Filamin protein fragments (Cher15-19wt, Cher15-19closedMSR, and Cher15-19openMSR) (Huelsmann et al., 2016) were cloned in pGTVL1-SGC vector (Structural Genomics Consortium, University of Oxford) using ligation-independent method (Gileadi et al., 2008) and expressed as Glutathione S-transferase (GST) fusion proteins in *Escherichia coli* BL21 Gold cells (Agilent Technologies) at 18°C for 22 h. The GST fusion proteins were purified with Protino Glutathione Agarose 4B (MachereyNagel), and the GST-tag was then cleaved using Tobacco Etch Virus (TEV) protease (Invitrogen, Life Technologies). The proteins were further purified by size exclusion chromatography (SEC) with a HiLoad 26/60 Superdex 75 column (GE Healthcare) in SEC buffer (20 mM Tris pH 7.5, 100 mM NaCl, 1 mM DTT) and concentrated using an Amicon ultracentrifugal 10K filter device (Merck Millipore). In this paper, the purified five-domain *Drosophila* Filamin fragments are called Filamin-wild type, Filamin-open and Filamin-closed. C-terminal fragments of Drak (360-532, 404-532, 435-532, and 458-532) cloned into pGTVL1-SGC were expressed at 20°C for 4 h. The cells were harvested by centrifugation at 6000 $g$ for 20 min at 4°C followed by cell resuspension in PBS (1 g of cells in 10 ml) and stored at −80°C.

### Binding assay

An aliquot (5 ml) of *E. coli* cells overexpressing Drak was thawed and supplemented with half a tablet of protease inhibitor (cOmplete ULTRA mini EDTA-free tablets, Roche Holding AG). Cell lysis was performed by sonication in three cycles, each consisting of 5 s on, 10 s off, at 10% amplitude. The lysate was then cleared by centrifugation at 35,000 $g$ for 30 min at 4°C and incubated with 900 µl of Protino Glutathione Agarose 4B (MachereyNagel) in PBS for 20 min on ice. The glutathione agarose was then washed three times with 5 ml of ice-cold PBS, using a centrifuge at 1000 $g$ for 2 min at 4°C. Next, the glutathione agarose was resuspended in an equal volume of ice-cold PBS, and 100 µl was transferred into a filter column. To probe the interaction, 100 µl (0.2-10 µM) of various Filamin fragments described above were incubated with the GST-Drak-Glutathione agarose for 20 min in 20 mM Tris-HCl pH 7.5, 100 mM NaCl and 1 mM DTT on ice. The agarose was then washed five times with 500 µl PBS, using a centrifuge at 1000 $g$ for 1 min at room temperature. Finally, bound proteins were eluted with 50 µl of 2× SDS-electrophoresis sample buffer and separated according to molecular weight using 12% Mini-PROTEAN TGX stain-free SDS-PAGE gels (Bio-Rad). Triplicate experiments were performed each time for every protein concentration. Total stain-free fluorescence intensity of Drak and Cheerio bands of each sample lane were quantitated using Imagelab software (Bio-Rad). The relative intensity of bound Filamin fragments was obtained by comparing the ratio of the intensity of Filamin to Drak intensity for each experiment.

### Yeast two-hybrid screen

To explore protein interacting with the open Filamin MSR, we sent a DNA fragment encoding domains 16-19 (aa 1518-1895 in the Genbank entry AAF55390.4) of *Drosophila* Filamin Cheerio to the Hybrigenics Services (Evry-Courcouronnes, France). This fragment contained mutations I1524E

and I1710E as the other Filamin-open constructs used in this study. The company cloned the fragment to the pB29 (N-bait-LexA-C fusion) yeast two-hybrid bait vector and conducted a yeast two-hybrid screen (Fields and Song, 1989) with *Drosophila* ovary library RP1 using 100 mM 3-aminotriazole to inhibit autoactivation of the bait. The screen resulted in 249 hits that were sequenced. Eleven of the hits corresponded to Drak fragments, and four of them were identical.

### Fly strains and upkeep

We used genomic GFP insertion of the *cher* locus, *cher*[WT GFP] (w;; P{ry[+t7.2]=neoFRT}82B GT{*cher*[GFP_mGFP6-2]}) and corresponding *cher*[closed MSR] and *cher*[open MSR] mutations, all expressing cher-GFP fusion proteins (Huelsmann et al., 2016). The *cher*[WT mCherry] was generated by integrating a *cher*-mCherry fusion construct to the same founder cher attP-insertion line (*cher*[s24]) that was used for the generation of *cher*-GFP lines (Huelsmann et al., 2016). The *spaghetti-squash* (*sqh*)-mCherry fly stock (#99923) carrying an extra genomic copy of *sqh* linked to mCherry in the second chromosome was obtained from the Bloomington *Drosophila* Stock Center. *Drak*[KO] line lacking most of the kinase domain was a gift from David R. Hipfner (Department of Medicine, University of Montreal, Canada) (Neubueser et al., 2010). The *Drak*-GFP allele was generated by creating a double-stranded DNA break 17 nucleotides before the *Drak* STOP codon and using a homologous recombination template in the pHD-sfGFP-ScarlessDsRed (Addgene) vector containing a red fluorescent protein marker induced in eye tissues. The template vector contained a 1 kb 5′ *Drak* genomic fragment linked to GFP with a 12 amino acid linker: GGSGGGSGGSGGS and a 2 kb 3′ *Drak* genomic fragment. The guide RNA target protospacer adjacent motif was mutated with a silent substitution in the homologous recombination template. The TH-attp40 *Drosophila* embryos carrying nanos-Cas9 in chromosome 2 were microinjected with the *Drak* guide RNA plasmid pCFD3 (Addgene) and the homologous recombination template plasmid at the University of Cambridge, Department of Genetics Microinjection service facility. The adults surviving after microinjection were crossed with w⁻ flies, and offspring with red fluorescent eyes were selected. A single fly line harboring *Drak*-GFP was verified by sequencing. Flies were maintained at 22°C on Nutri-Fly Bloomington media (Genesee Scientific).

### Embryo preparation and imaging in cellularization

For embryo collection, flies were kept on grape agar dishes (FlyStuff Grape Agar Premix, Genesee Scientific) with a drop of yeast. Dishes were changed two to three times daily for 2-3 days. Embryos were collected after allowing the flies to lay eggs for 2-3 h. Halocarbon oil 27 (Sigma-Aldrich) was added to the agar plate to better visualize the embryos. Embryos close to final nuclear divisions were picked, and excess oil was removed. Embryos were dechorionated using household bleach for 1.5 min and rinsed in three drops of water. Embryos were placed on a 35 mm high glass-bottom dish (ibidi) in PBS, 0.1 mM ascorbic acid.

Imaging was carried out using a Leica SPX8 Falcon confocal microscope with an HC PL APO 63×/1.20 water immersion objective. For actomyosin ring dynamics study, embryos were imaged at 3-min intervals over a $20×20×40 \ \mu m^3$ 3D section with a voxel size of $0.08×0.08×0.35 \ \mu m^3$. For localization studies, the time interval was 90 s, with dimensions of $20.5×20.5×30.8 \ \mu m^3$ and a voxel size of $0.16×0.16×0.50 \ \mu m^3$. In the colocalization studies, pixel size was $0.14×0.14 \ \mu m^2$.

### Pupal preparation and imaging of developing flight muscles

Flies were transferred to fresh vials and kept for 6-8 days until white prepupae appeared. Prepupae were collected and aged for 20-24 h. Pupae were secured on a double-sided tape, and the dorsal side of the pupal case was removed near the head and thorax using a fine needle. Pupae were detached from the tape using water, dried, and arranged in a row on a 35 mm glass-bottom dish, with the dorsal abdomen adhered to double-sided tape and the thorax towards the glass. The pupae were firmly secured in place by placing a tape on top of them and tightened until thoraces were pressed against the glass. For high-resolution imaging, a small drop of halocarbon oil 27 was added before mounting each pupa.

Low-resolution images were acquired using a Nikon A1R laser scanning confocal microscope with a Plan Apo 20× air objective. Imaging dimensions

were 570×570×34 μm³ with a voxel size of 0.56×0.56×2 μm³, captured at 1-h intervals. High-resolution images were obtained using the same microscope with a Plan Apo 60× oil objective or a Leica SPX8 Falcon confocal microscope with an HC Plan Apo 63× glycerol objective. For the Nikon, high-resolution imaging dimensions were 100×100×15 μm³ with a voxel size of 0.099×0.099×3 μm³. For the Leica, dimensions were 92×92×3 μm³ with a voxel size of 0.075×0.075×3 μm³. The time interval for both setups was 1 h. The pupa IFM development movie was done with 0.099×0.099 μm² pixel size and 1 min time interval. Colocalization studies were done using Leica with pixel size of 0.14×0.14 μm².

### Adult flight muscle preparation and imaging

IFMs were dissected from adult flies 1-2 days after eclosion. After removing the head and legs, the thorax was gently punctured with spring scissors. Thoraces were fixed in 4% paraformaldehyde PBS for 30 min at room temperature and then washed twice using PBS with 0.1% Triton X-100 for 5 min each. The thorax was cut in half with a longitudinal incision. The abdomen was removed, and the half thoraces were stained with 0.5 U/ml Alexa Fluor 555 phalloidin (Thermo Fisher Scientific) overnight at 6°C.

The half-thorax was submerged in a drop of Vectashield Vibrance (VectorLabs) mounting medium on a microscope slide, and the flight muscles were dissected using a fine tungsten needle (Fine Science Tools). A coverslip was placed on the sample and sealed with nail polish. Muscles were imaged using a Nikon A1R confocal microscope with a 60× oil objective. For each muscle sample, we captured two to three images covering a 50×50 μm² area of muscle attachment sites, yielding approximately 20-30 individual sites for measurement. The Filamin-GFP signal was used to confirm the presence of intact attachment sites.

### Western blotting

Embryos were collected approximately half an hour after final nuclear division into 1× Laemmli sample buffer containing 2-mercaptoethanol, 1 mM vanadate, 10 mM sodium fluoride, and 2 mM beta-glycerophosphate to a concentration of 2.5 embryos/μl. All embryo samples were prepared in technical duplicates. Phosphorylated Sqh was detected with 1:1000 dilution of Anti-Myosin pS19/pS20 (Rabbit) antibody (#600-401-416, Rockland). To quantify total Spaghetti Squash (Sqh), a custom-ordered Sqh antibody from Boster Biological Technology CO., Ltd. Full-length Sqh (Uniprot ID: P40423) expressed in *E. coli* was used for rabbit immunization and for affinity purification. Sqh antibody was used at a 1:3000 dilution. The secondary antibody, Anti-Rabbit HRP (#ab6721, Abcam), was applied at a 1:3000 dilution for phosphorylated Sqh detection and at a 1:6000 dilution for total Sqh. Chemiluminescent detection was performed using SuperSignal West Pico PLUS substrate (Thermo Fisher Scientific). Samples were run on 12% stain-free gels (Bio-Rad), and proteins were transferred onto a 0.2 μm PVDF membrane using the Trans-Blot Turbo Transfer Pack kit (Bio-Rad). The transfer was carried out using the Trans-Blot Turbo Transfer System (Bio-Rad) for 5 min at 1.3 A and 25 V. Images were analyzed using Image-Lab software (Bio-Rad). Lane and band background were subtracted with rolling ball method using the suggested default settings. Stain-free total protein signal was used as the loading control. To compare the relative changes of phosphorylated myosin and total myosin, fold change was calculated by normalizing to the average of wild-type mean, and the comparisons were presented as the ratio of the fold changes.

### Image analysis

Confocal images of cellularization were analyzed using Fiji (Schindelin et al., 2012). XY projections of the cortical region were obtained by taking five slices starting 1 μm from the edge of the embryo. For deeper regions within the embryo, XY projections were created by summing three slices around the actomyosin ring. A Gaussian blur with sigma=1 was applied. Minimum and maximum gray values were set consistently for each time point. XZ projections were generated by using the 'Reslice' function over a 7.5 μm thick area across three nuclei using 15 slices and creating a sum projection and applying Gaussian blur with sigma=2.

Cortical intensity change over time was measured as the mean intensity of seven slices starting from the edge of the embryo. This was done in each time point for each fluorophore, and background level was measured from samples without the corresponding fluorophore. Background was corrected in each time point, and each sample was min-max normalized.

Vertical fluorescence intensity profiles during cellularization were generated by extracting intensity data from each plane of the 3D image. The shape of the actomyosin ring was analyzed using Filamin-GFP signal at the peak position of the vertical intensity profiles. The first frame after the 14th nuclear division was set as the zero-time position. Samples that showed no contraction after passing the basal level of the nuclei were excluded. Noise was reduced using the Noise2Void plugin (Krull et al., 2019 preprint) with a model trained on approximately ten projections of the ring per genotype. The slice was thresholded using the Huang method (Huang and Wang, 1995) and smoothed with the Shape Smoothing plugin (https://github.com/thorstenwagner/ij-shape-smoothing) (relative Fourier descriptors=8). Circularity was measured for each ring by measuring the outlines of the inner edge of the rings.

The relationship between circularity and time in cellularization for each genotype was fitted using generalized additive mixed model using the gamm() function from the R mgcv package (https://CRAN.R-project.org/package=mgcv). Average circularity was calculated in each time point, and, to account for non-linear trends in circularity over time, we used a smooth term for time, s(Time). We included a random intercept for each sample to account for baseline differences in circularity across samples. The final model had the following form:

$$\text{gamm}(\text{Circularity} \tilde{~} s(\text{Time}), \text{random} = \text{list}(\text{Sample} = \tilde{~}1), \text{data} = \text{genotype\_data}).$$

Low-resolution images of the pupal IFMs (Fig. 4A) were created by sum projecting four imaging planes. High-resolution images (Fig. 4B,C) were obtained from single slices, and a Gaussian blur with sigma=1 was applied.

Changes in Drak intensity in developing IFMs were measured from a 20 μm-thick section selected at the middle of the muscle. Using 'Reslice' function, this section was computationally reoriented so that ten longitudinal slices could be obtained across the myotubes and tendons. These slices were then z-projected using the 'sum slices' option. The mean intensity of the Drak-GFP signal in the visible cytoplasmic area was measured. Background was subtracted by taking the average intensity of control samples at each time point, and each sample was min-max normalized. Displacement of the myotubes was measured by tracking the x-coordinate of the tendon-myotube junction and subtracting these coordinates from the first time point. Single dorsal-longitudinal IFMs from three pupae were analyzed. Time points between samples were aligned by adjusting the time axis so that the minimum x-coordinates matched, and the first time point was set to 24 h APF for all samples, corresponding to when Drak intensity begins to increase. Normalized Drak intensity and displacement values were plotted to show the relationship between Drak intensity and myotube compaction and elongation, and both were fitted with a local regression model in R.

Muscle attachment site images for measurement were processed as follows. For the GFP channel, a Gaussian blur with sigma=1 was applied, followed by contrast enhancement with 0.1% saturated pixels. For the phalloidin channel, a Gaussian blur with sigma=2 was applied, and contrast was enhanced with 1% saturated pixels. To further enhance edge features, a Laplacian filter with smoothing set to 10 using the FeatureJ plugin (https://imagescience.org/meijering/software/featurej/) was applied. Attachment site length was manually measured from the inner side of the terminal Z-line to the tip of the myofiber using the line selection tool.

To compare the muscle attachment site length in different fly genotypes, we fitted a generalized linear mixed model, using function glmer() from the lme4 R package (https://CRAN.R-project.org/package=lme4). The response variable, attachment site length, was modelled with a gamma distribution and a log link function to account for positively skewed distribution. Genotype was included as a fixed effect to estimate how attachment site length differed among genotypes. To account for non-independence of images from the same sample, we included a nested random intercept for image number within each sample, allowing for baseline variation among samples and their respective images. The final model was specified as follows:

glmer(att_site_length~genotype+(1|sample/image_number),
data=att_site_length_d.f., family=Gamma(link="log")).

To evaluate genotype effects, custom contrasts were specified to test three comparisons: (1) wild type versus Filamin-closed, (2) wild type versus

Drak-KO, and (3) double mutant (Drak-KO, Filamin-closed) versus the additive effects of the individual Filamin-closed and Drak-KO genotypes. *P*-values were computed using two-sided Wald z-tests for the specified contrasts from the fitted model, using model-based standard errors, and adjusted with Bonferroni correction.

To quantify colocalization for Drak-GFP and Filamin-mCherry during the cellularization priming phase and tendon-myotube attachment site maturation, we measured thresholded Manders' colocalization coefficients (M1 and M2) and the tPCC. Single-fluorophore flies were imaged with the same settings as double-fluorophore flies to determine the background levels of each channel. Global threshold was set to the weighted mean plus three standard deviations. M1 is the sum of above-threshold Drak-GFP pixel intensity values that overlap with above-threshold Filamin-mCherry signal, divided by the sum of above-threshold Drak-GFP pixel intensity values. M2 is the analogous value for Filamin-mCherry intensities. tPCC is the linear intensity correlation coefficient between Drak-GFP and Filamin-mCherry channels calculated from the above-threshold pixel intensity values. M1, M2, and tPCC were calculated separately for each image. Sample-level means and standard deviations were weighted by the number of informative pixels in each image: M1 by the count of Drak pixels above threshold, M2 by the count of above-threshold Filamin pixels, and tPCC by the count of pixels that exceeded the threshold in both channels. Population-level summaries and one-sample *t*-tests were performed on the sample mean values. For tPCC, sample means were Fisher z-transformed prior to the *t*-test and confidence interval estimation, and the resulting estimates were back-transformed to the correlation scale.

As image areas with either no expression of the fluorophores or with high autofluorescence can skew colocalization analysis, we created an ImageJ macro to define regions of interest (ROIs) for the embryo colocalization analysis. The embryo edge areas that had high autofluorescence were manually excluded. Nuclear areas were excluded from the ROI as both fluorophores were mostly absent inside nuclei as expected. The ROI surrounding the nuclei was segmented by using Filamin-mCherry signal only as it defined the boundaries more consistently. The background value was first subtracted, then mean filter was applied with radius of 4 px and finally using Auto Threshold with method 'Otsu'. Additionally, the imaging area had some high-intensity autofluorescent spots in both channels that were automatically excluded. Spot masks for each channel were created by first subtracting background with rolling ball method using 50 px radius. Then, mean filter with radius of 3 px was applied, and the resulting image was subtracted from the background-removed image. Finally, Auto Threshold with method 'MaxEntropy' was used to segment the bright spots, and the spots were dilated once with run("Dilate"). The spot mask was then excluded from the final ROI. The ROIs for tendon-myotube attachment site samples were defined manually by including the visible tendon signal and avoiding myotube tissue and highly autofluorescent areas. Representative colocalization images and ROI masks of each sample studied are given in Figs S3 and S4.

## Use of AI tools

AI tools (ChatGPT v4.0, v4.1, and v5.0, OpenAI) were used to improve ImageJ and R code, and to rephrase individual sentences to increase clarity and check for grammatical errors.

## Acknowledgements
We thank Sven Huelsmann for designing the yeast two-hybrid bait and for preparing the cher-GFP and cher-mCherry fly lines when working for the project. Petri Papponen is acknowledged, for general laboratory help and guidance; Sara Calhim for help with statistics; Iida Koskela for developing the Drak-Filamin binding assay, Mikko Hakanen for working on image analysis and preliminary results on cellularization dynamics; Lassi Paavolainen for helping with colocalization analysis; and Visa Ruokolainen for all the help and guidance with confocal imaging. We thank David R. Hipfner for the Drak-KO fly stock; Nick Brown for collaboration and sharing facilities during the early phase of the work; Stefano De Renzis and Daniel Krueger for introduction to the fly cellularization imaging; and University of Cambridge, Department of Genetics, fly facility, for the microinjection service..

## Competing interests
The authors declare no competing or financial interests.

## Author contributions
Conceptualization: R.O.K., C.T., J. Y.; Data curation: R.O.K., C.T., J. Y.; Formal analysis: R.O.K., C.T., J. Y.; Funding acquisition: J.Y.; Investigation: R.O.K., C.T., H.J.G., J.Y.; Methodology: R.O.K., C.T., H.J.G., J.Y.; Project administration: R.O.K., J.Y.; Visualization: R.O.K., C.T., J.Y.; Writing – original draft: R.O.K., C.T., J.Y.; Writing – review & editing: R.O.K., C.T., H.J.G., J.Y

## Funding
This work was funded by Research Council of Finland grant 343444 to J.Y.; and by the Department of Biological and Environmental Science and the Doctoral Programme in Biological and Environmental Science, Jyväskylän Yliopisto. Open Access funding provided by Jyväskylän Yliopisto. Deposited in PMC for immediate release.

## Data and resource availability
The confocal images and the ImageJ and R code used for analysis are available at Fairdata repository's Etsin service (doi:10.23729/1899cb2a-1647-4b2f-b20c-aed491ce551b). All other relevant data and details of resources can be found within the article and its supplementary information.

## Peer review history
The peer review history is available online at https://journals.biologists.com/bio/lookup/doi/10.1242/bio.062185.reviewer-comments.pdf

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
