## [Peer Review File · Biology Open]

Drak is a potential binding partner of Drosophila Filamin

Chandan Thapa, Jari P. Yläanne, Hannah J. Green and Riku Oskari Korkiamäki

DOI: 10.1242/bio.062185

Editor: Cathy Jackson

Review timeline

Submission to sister journal:	31 January 2025
Editorial decision at sister journal:	3 March 2025
Transfer to Biology Open:	1 August 2025
Editorial decision:	9 September 2025
First revision received:	12 January 2026
Editorial decision:	16 February 2026
Second revision received:	20 March 2026
Accepted:	24 March 2026

Original submission to sister journal

MS Title: Drak is a potential binding partner of Drosophila Filamin

Authors: Riku Oskari Korkiamäki; Chandan Thapa; Jari P Yläanne; Hannah J Green

We have now reached a decision on the above manuscript.

To see the reviewers' reports and a copy of this decision letter, please go to:

As you will see from their reports, the reviewers raise a number of substantial criticisms that prevent me from accepting your paper for publication. Some of the analysis presented in the current version of the manuscript is preliminary and requires more work beyond the scope of the time available for revisions.

I am very sorry to give you such disappointing news, but we are currently under great pressure for space and it takes a very enthusiastic recommendation by the referees for a manuscript to be accepted.

I do hope you find the comments of the reviewers helpful in allowing you to revise the manuscript for submission elsewhere,

Reviewer 1

Comments for the author

This paper attempts to identify contributors to Filamin mechanical signal transduction. An open form of Filamin, the form known to be generated by mechanical force, is shown to interact with fragments of the C-terminal half of Drak in yeast two hybrid and GST pull down assays. In both the Drosophila syncytium and Drosophila muscle cells, localizations and loss-of-function effects of Filamin and Drak are compared.

The paper contains a well-written review of the literature, intriguing biochemically data, development of new reagents, and careful discussions of caveats that prevent clear conclusions. However, it is problematic that the current results do not allow the authors to make clear conclusions. In my view, there is insufficient conceptual advance.

The yeast two-hybrid and biochemical results are striking, but they only show that a portion of Drak interacts with the open form of Filamin. It is possible that the full length Drak protein does not interact with Filamin.

In two distinct in vivo contexts, close colocalization of Drak and Filamin expected for a direct interaction was not evident. For example, either one or both of the proteins displayed punctate distributions in the contexts (Fig 2D in the X-Y sections; Fig 4B-C at 37h) but puncta-to-puncta colocalization of Drak and Filamin is not evident/reported.

In two distinct in vivo contexts, mutants of a closed form of Filamin, that did not interact with Drak in vitro, did not mimic the effects of a Drak loss of function mutant (Fig 3; Fig 4E-F). The single mutants actually displayed opposite abnormalities in muscle cells (Fig 4F), inconsistent with them working together positively through the implicated interaction. Although there is some evidence of a genetic interaction affecting muscle cells, the data do not support a model of Filamin recruiting Drak to cause a functional outcome and an alternate model is unclear.

More minor concerns:

- the Filamin-mCherry does not seem to be explained in the material and methods section
- unclear how the separately collected Sqh-mCherry data was temporally aligned with the co-collected Drak-GFP/Filamin-mCherry data for all to start at 0s in the graph in 2B
- in the Figure 2 legend, "Squash" should be replaced by "Spaghetti squash".

Reviewer 2

Comments for the author

SUMMARY OF THE ADVANCE MADE IN THIS PAPER AND ITS POTENTIAL SIGNIFICANCE TO THE FIELD

The advance in this paper is the demonstration that at the beginning of cellularization, Drak (a death-associated protein that acts as a myosin light chain kinase) binds with activated Filamin (*Drosophila* Cheerio, cher), an actin-binding and mechanosensory protein during the initiation of microfilament ring constriction. Furthermore, Filamin and Drak interact with the development of indirect flight muscles, showing a genetic effect.

This article provides a brief study that ties together Filamin and Drak together in two distinct biological processes. Drak is involved in the assembly of microfilament rings during cellularization, and the authors discovered that Filamin has role in activating Drak. The authors also discovered that Filamin and Drak were involved in the indirect flight muscles and showed a genetic interaction between the genes.

SUGGESTIONS TO AUTHORS

- (1) Wild-Type Filamin shows a reproducible weaker interaction than Open Filamin (Closed Filamin did not show an interaction). An accurate Kd could not be determined, and the range was within almost two orders of magnitude. The authors should collect more data to narrow this large range.
- (2) In the text in the first paragraph "Drak and Filamin *Drosophila* flight muscle development," the callout for Fig. 4D came before Fig. 4C. This confused me and interrupted my reading. Change the order to coincide with the text.
- (3) Likewise, the layout of Fig. 4 is jumbled. It would be helpful to see the panels in sequence.
- (4) There are some grammatical errors that should be fixed.

(5) The text for the Results is quite brief and there are only four Figures. It seems that this Research Article would fit better as a Short Report.

Reviewer 3

Comments for the author

SUMMARY OF THE ADVANCE MADE IN THIS PAPER AND ITS POTENTIAL SIGNIFICANCE TO THE FIELD

This interesting study seeks to understand the physical and functional relationship between Filamin and Drak using biochemical, genetic, and imaging approaches. Although some of the data are convincing, there are many gaps in the experimental descriptions that make it difficult to evaluate. Even if those gaps were filled, the manuscript seems thin for a research article and would benefit from more investigation into the functional consequences of the interaction.

SUGGESTIONS TO AUTHORS

Figure 1:

A schematic of Filamin would be helpful.

P10 line 53-55 Please be more specific about the short GST-Drak fragment and the five domain Filamin fragments

For the expt in Fig 1B, more detail is needed in the text and in the legend to describe the experiment.

For the expt in Fig 1C, more detail is needed. For example, what specifically is interacting with Filamin? Is the black line actual data, or just a model? I don't think it's necessary to establish the K_d , so if you can't do it I would leave it out.

Figure 2:

Page 11, lines 18-28. I'm not seeing the features that you're describing in the images including transient localization to the metaphase or cellularization furrows. It's hard to appreciate the relative positions of Drak, Sqh, and Filamin in Fig 2A. Showing (1) a representative merged image for Drak and Filamin, and (2) combining Drak-GFP and Sqh-mCherry to show the overlap directly would solve the problem of their relative positions.

Fig 2D- the label on the figure indicates "front distance from edge". Do you mean the surface of the embryo? Please clarify. For the intensity plots, I see the very small Drak peak described, but to be convinced the n should be higher and some statistical analysis used.

Page 11, lines 31-32. "Drak-GFP became mainly diffuse in the cytoplasm". From the images it doesn't seem different than it was earlier in development.

Figure 3:

Page 11, lines 40-50. This section is hard to follow and I'm not sure I understand the setup of the experiment. Are you overexpressing the Filamin-closed MSR mutant and simultaneously expressing wild type Filamin-GFP? I don't think I understand the experiment well enough to evaluate the results and conclusions.

Figure 4:

Need a cartoon to depict the overall tissue organization/cell types as without that background this section is very hard to follow. This makes it very difficult to evaluate. For Fig 4E, insets with significantly higher magnification should be included to illustrate the attachment site phenotype.

Discussion:

Page 12 line 35. It seems like this is the first time that MSR is defined though the abbreviate is used throughout.

Page 13 line 15-18. How are you defining the strength of the interaction? Referring to the specific figures will help the reader.

Page 13-14. The authors tested the hypothesis that Filamin and Drak would function together in cellularization but got negative results. No alternative explanations are provided. It is also not

clear why the authors were unable to reproduce published findings. This raises questions about how the experiments were done.

Transfer to Biology Open

Author response to previous reviewers' comments

Korkiamäki et al: Drak is a potential mechanotransduction partner of Filamin

Explanation of major changes and comments to the reviewers

According to the reviewers' suggestions, the major changes in the current manuscript are:

1. Explanatory figure is added to the introduction explaining the Filamin protein structure and two developmental events studied: cellularization and indirect flight muscle formation
2. Extensive colocalization analysis between Drak-GFP and Filamin-mCherry is added in the two developmental events: the priming phase of cellularization (Current Figure 4, Supplemental Figure S2 and Supplemental Table S1) and the tendon-myotube attachment site maturation stage of indirect flight muscle development (Current Figure 6, Supplemental Figure S3 and Supplemental Table S2)
3. To clarify the presentation, former Figure 4 has now been splitted to two separate figure (Current Figure 7 and 8)

Below we give detailed replies to reviewers's comments

Comments from the Reviewers:

Reviewer 1: This paper attempts to identify contributors to Filamin mechanical signal transduction. An open form of Filamin, the form known to be generated by mechanical force, is shown to interact with fragments of the C-terminal half of Drak in yeast two hybrid and GST pull down assays. In both the Drosophila syncytium and Drosophila muscle cells, localizations and loss-of-function effects of Filamin and Drak are compared.

The paper contains a well-written review of the literature, intriguing biochemically data, development of new reagents, and careful discussions of caveats that prevent clear conclusions. However, it is problematic that the current results do not allow the authors to make clear conclusions. In my view, there is insufficient conceptual advance.

The yeast two-hybrid and biochemical results are striking, but they only show that a portion of Drak interacts with the open form of Filamin. It is possible that the full length Drak protein does not interact with Filamin.

Authors' response

We thank the reviewer for the comment, and we partially agree with it: we have not been able to show biochemical interaction with full length filamin and full length Drak. We were not able to purify full length Drak for biochemical experiment. However, in the revised manuscript we show that Filamin and Drak colocalize at specific stages of two developmental processes in vivo; During the priming phase of embryo cellularization (Fig. 4, revised manuscript) and during the indirect flight muscle tendon-myotube attachment site maturation phase (Fig. 6, revised manuscript).

In two distinct in vivo contexts, close colocalization of Drak and Filamin expected for a direct interaction was not evident. For example, either one or both of the proteins displayed punctate distributions in the contexts (Fig 2D in the X-Y sections; Fig 4B-C at 37h) but puncta-to-puncta colocalization of Drak and Filamin is not evident/reported.

Authors' response

According to reviewer's suggestion we have now added extensive colocalization analysis. In our original confocal timelapses the Drak-GFP signal was too weak to allow colocalization analysis, but now we have collected new images from several samples with higher laser power and analyzed colocalization between Drak-GFP and Filamin-mCherry. The analysis shows extensive colocalization and moderate intensity correlation.

In two distinct *in vivo* contexts, mutants of a closed form of Filamin, that did not interact with Drak *in vitro*, did not mimic the effects of a Drak loss of function mutant (Fig 3; Fig 4E-F). The single mutants actually displayed opposite abnormalities in muscle cells (Fig 4F), inconsistent with them working together positively through the implicated interaction. Although there is some evidence of a genetic interaction affecting muscle cells, the data do not support a model of Filamin recruiting Drak to cause a functional outcome and an alternate model is unclear.

Authors' response

Yes, we agree with the reviewer, this is a weak point in our model. However, we think that the new colocalization data supports the model of direct interaction in both developmental contexts studied. In the current manuscript version, we also explain why stronger Filamin mutants could not be used in these analyses. The epigenetic effect of Drak-KO and Filamin-closed mutation in muscle attachment site development does not prove direct interaction, rather it suggests that both gene products are involved in the same event. However, we point out that statistically, the changes of the double mutant were different than the additive changes of the single mutants. This suggests that the two proteins may have some common signaling functions in addition to their independent functions.

More minor concerns:

-the Filamin-mCherry does not seem to be explained in the material and methods section - unclear how the separately collected Sqh-mCherry data was temporally aligned with the co-collected Drak-GFP/Filamin-mCherry data for all to start at 0s in the graph in 2B -in the Figure 2 legend, "Squash" should be replaced by "Spaghetti squash".

Authors' response

We added explanation to Filamin-mCherry (cher-mCherry) fly line to the Methods section. Figure legend in Fig 2 (Fig 3 in the revised manuscript) states how samples were aligned temporarily: "Distribution of Drak-GFP, (top panel), Filamin -mCherry (middle panel), and Sqh-mCherry (bottom panel) during the first 0-21 minutes starting from the last nuclear division." We changed "Squash" to "Spaghetti squash".

Reviewer 2: SUMMARY OF THE ADVANCE MADE IN THIS PAPER AND ITS POTENTIAL SIGNIFICANCE TO THE FIELD

The advance in this paper is the demonstration that at the beginning of cellularization, Drak (a death-associated protein that acts as a myosin light chain kinase) binds with activated Filamin (*Drosophila* Cheerio, *cher*), an actin-binding and mechanosensory protein during the initiation of microfilament ring constriction. Furthermore, Filamin and Drak interact with the development of indirect flight muscles, showing a genetic effect.

This article provides a brief study that ties together Filamin and Drak together in two distinct biological processes. Drak is involved in the assembly of microfilament rings during cellularization, and the authors discovered that Filamin has role in activating Drak. The authors also discovered that Filamin and Drak were involved in the indirect flight muscles and showed a genetic interaction between the genes.

SUGGESTIONS TO AUTHORS

(1) Wild-Type Filamin shows a reproducible weaker interaction than Open Filamin (Closed Filamin did not show an interaction). An accurate K_d could not be determined, and the range was within

almost two orders of magnitude. The authors should collect more data to narrow this large range.

Authors' response

As suggested by reviewer 3, we removed the Kd calculation. We agree with the reviewer that the experimental variation is too large for this analysis. However, we point out that despite that we can clearly demonstrate that wild type Filamin fragment binds to Drak weaker than the open Filamin and that closed filamin does not bind.

(2) In the text in the first paragraph "Drak and Filamin Drosophila flight muscle development," the callout for Fig. 4D came before Fig. 4C. This confused me and interrupted my reading. Change the order to coincide with the text.

(3) Likewise, the layout of Fig. 4 is jumbled. It would be helpful to see the panels in sequence.

(4) There are some grammatical errors that should be fixed.

(5) The text for the Results is quite brief and there are only four Figures. It seems that this Research Article would fit better as a Short Report.

Authors' response

We thank the reviewer for this suggestion. To clarify the presentation, we decided to divide Fig 4 into two parts. The first part (Fig. 4A-D) about pupal flight muscles is now Fig 7A-D on the revised manuscript, and the second part (Fig 4.F-E) is now Fig. 8A-B. We decided to keep the original panel order in Fig. 7A-D as moving the graph after 7A would impede the comparison between the localization of different proteins. In the text, we now first refer to Drak-related panels Fig. 7A,B,D in one callout.

Reviewer 3: SUMMARY OF THE ADVANCE MADE IN THIS PAPER AND ITS POTENTIAL SIGNIFICANCE TO THE FIELD

This interesting study seeks to understand the physical and functional relationship between Filamin and Drak using biochemical, genetic, and imaging approaches. Although some of the data are convincing, there are many gaps in the experimental descriptions that make it difficult to evaluate. Even if those gaps were filled, the manuscript seems thin for a research article and would benefit from more investigation into the functional consequences of the interaction.

Authors' response

We added colocalization analysis to show more convincing evidence of interaction in vivo in cellularization priming phase (Fig. 4, revised manuscript) and indirect flight muscles (Fig. 6, revised manuscript) myotube-tendon attachment site maturation

SUGGESTIONS TO AUTHORS

Figure 1:

A schematic of Filamin would be helpful.

P10 line 53-55 Please be more specific about the short GST-Drak fragment and the five domain Filamin fragments:

Authors' response

We added an introductory figure (Fig. 1, revised manuscript) that shows Filamin structure as well as introduces to cellularization and indirect flight muscle development. Y2H filamin fragment domains 16-19 were specified on line 48 on the same page as "aa 1518-19895". There was however a typo that we corrected to "aa 1518-1895".

For the expt in Fig 1B, more detail is needed in the text and in the legend to describe the experiment.

For the expt in Fig 1C, more detail is needed. For example, what specifically is interacting with Filamin? Is the black line actual data, or just a model? I don't think it's necessary to establish the Kd, so if you can't do it I would leave it out.

Authors' response

We added more details in the text and legend to describe Fig 1 (Fig. 2 in revised manuscript). We removed Kd calculation. The referee was right, the lines in Fig 1C (Fig. 2C, revised manuscript) showed the means of each data point instead of the hyperbolic model as described in the legend.

Figure 2:

Page 11, lines 18-28. I'm not seeing the features that you're describing in the images including transient localization to the metaphase or cellularization furrows. It's hard to appreciate the relative positions of Drak, Sqh, and Filamin in Fig 2A. Showing (1) a representative merged image for Drak and Filamin, and (2) combining Drak-GFP and Sqh-mCherry to show the overlap directly would solve the problem of their relative positions.

Fig 2D- the label on the figure indicates "front distance from edge". Do you mean the surface of the embryo? Please clarify. For the intensity plots, I see the very small Drak peak described, but to be convinced the n should be higher and some statistical analysis used.

Page 11, lines 31-32. "Drak-GFP became mainly diffuse in the cytoplasm". From the images it doesn't seem different than it was earlier in development.

Authors' response

According to reviewer's suggestion, we added references to the text for timepoint where the transient localization to the metaphase furrows (Fig. 3A 0 min and 3 min, revised manuscript) and forming cellularization furrows (Fig. 3A 6 min and 9 min). We added a new figure showing colocalization analysis that corresponds to time point 6-9 min, where the overlap between Drak and Filamin is shown (Fig. 4, revised manuscript). We removed the label "front distance from edge" and instead explained in figure legend that the distance was measured from the surface of the embryo.

Regarding the localization of Drak at the cellularization front during the later stages of cellularization, we agree with the reviewer that the signal is weak, and we rather emphasize the localization at the priming phase. We have changed in the text: "Drak-GFP became mainly diffuse to the cytoplasm" to "Drak remained mainly diffuse in the cytoplasm."

Figure 3:

Page 11, lines 40-50. This section is hard to follow and I'm not sure I understand the setup of the experiment. Are you overexpressing the Filamin-closed MSR mutant and simultaneously expressing wild type Filamin-GFP? I don't think I understand the experiment well enough to evaluate the results and conclusions.

Authors' response

We thank the reviewer for pointing out the unclear parts. The Filamin (gene *cher* on *Drosophila*) constructs we used are all genomic insertions to the *cher* locus and as such are under native promoters and not overexpressed. We added explanation to methods-section that each Filamin construct contains a GFP tag.

Figure 4:

Need a cartoon to depict the overall tissue organization/cell types as without that background this section is very hard to follow. This makes it very difficult to evaluate. For Fig 4E, insets with significantly higher magnification should be included to illustrate the attachment site phenotype.

Authors' response

According to reviewer's suggestion, we have added a cartoon about IFM development in the introduction section (Current Figure 1D). Unfortunately, we do not have EM images of these mutant muscle attachment site at the moment, but we refer to our previous paper (Green et al, 2018).

Discussion:

Page 12 line 35. It seems like this is the first time that MSR is defined though the abbreviate is used throughout.

Authors' response

We have checked to use of abbreviations

Page 13 line 15-18. How are you defining the strength of the interaction? Referring to the specific figures will help the reader.

Authors' response

The referee is right, since we did not get the Kd, we cannot evaluate the strength of the interaction. In the revised manuscript only compare the binding between different Filamin mutants qualitatively.

Page 13-14. The authors tested the hypothesis that Filamin and Drak would function together in cellularization but got negative results. No alternative explanations are provided. It is also not clear why the authors were unable to reproduce published findings. This raises questions about how the experiments were done.

Authors' response

We think that there has been a misunderstanding here. As far as we know, Filamin closed mutant has not been tested in this context before. We were however able to replicate previous results of Drak-KO effect on cellularization actomyosin ring circularity and phosphorylation levels of myosin as reported on the reference Chougule et. al., 2016. As per the reviewer's suggestion, we now have added a section to discussion why we did not observe similar effects on Filamin closed mutant.

First decision letter

MS ID#: bio.062185

MS Title: Drak is a potential binding partner of Drosophila Filamin

Authors: Riku Oskari Korkiamäki; Chandan Thapa; Jari P Yläanne; Hannah J Green

I have now reached a decision on the above manuscript.

The reviewer reports are shown at the bottom of this email or can be accessed, together with a copy of this decision letter, by going to:

As you will see, the reviewers raised a number of substantial criticisms that prevent me from accepting the paper at this stage.

They suggest, however, that a revised version might prove acceptable, if you can address their concerns. If you think that you can deal satisfactorily with the criticisms on revision, I would be pleased to see a revised manuscript. We would then return it to the reviewers.

At this stage, we also ask you to ensure your manuscript complies with our formatting guidelines. Provided you are able to fully address the referees' comments, we are positive about publication of your paper (we accept over 95% of revision submissions) and therefore hope you won't mind any extra work involved in reformatting your manuscript at this point.

Please upload both a 'clean' version of your Word file, along with a highlighted version clearly showing where you have made changes in the revised manuscript. Please avoid using 'Track changes' in Word files as these are lost in PDF conversion.

I should be grateful if you would also provide a point-by-point response detailing how you have dealt with the points raised by the reviewers in the 'Response to Reviewers' box. Please attend to all of the reviewers' comments. If you do not agree with any of their criticisms or suggestions please explain clearly why this is so.

Reviewer 1

Comments for the author

require clarification or strengthening before publication. In particular, the authors should (i) address the apparent discrepancy between the in vitro binding assays and the in vivo phenotypic effects of the Filamin-closed mutant during cellularization, and (ii) provide a more clear interpretation of Drak's functional contribution in muscle development, given the largely diffuse localization at later pupal stages. Some concerns are listed below:

Major comments:

1. Study lacks a potential mechanism by which Filamin regulates Drak kinase activity in response to mechanotransduction.
2. Authors show that Drak and Filamin colocalize during furrow initiation. However, the Filamin closed MSR mutant does not phenocopy loss of Drak. This weakens the argument that Filamin is essential for Drak recruitment or activity in embryos. Authors should consider redundancy or partial function of the closed-MSR mutant. Along those lines in figure 5, graph in figure 5b doesn't reflect the data in figure 5a with regards to the circularity differences seen between Drak KO vs WT and Filamin mutant. Given that the amount of total myosin is variable between samples, loading control should be included and amount of pMyosin should be represented as ratio of pMyo/total myosin.
3. In figures 6 and 7, authors suggest that Drak and Filamin co-localize at tendons. However, tendons and myotubes should be stained for using established markers.
4. In figure 8, the phenotypic readout (length of attachment sites) is relatively coarse. It would strengthen the argument if additional readouts (e.g. sarcomere integrity) were included to assess the physiological consequences.
5. Throughout the paper, authors rely heavily on colocalization analyses using Manders' coefficients and Pearson correlations. While helpful, the sample sizes are limited, and error/variance estimates are sometimes broad. More replicates, and statistical details (e.g. exact p-values, n per image per embryo/pupa) should be presented to solidify conclusions.

Reviewer 2

Comments for the author

In their recent paper, Korkiamäki et al. show that myosin light chain kinase Drak biochemically interacts with Filamin, an actin-crosslinking protein that is conserved from flies to humans. Interestingly, the authors show that Drak interacted with the mechanosensitive domain of Filamin, when it is in its open form. The same domain of wild-type Filamin, as well of a mutant form in the closed state, did not, or only weakly, interact with recombinant Drak. Using GFP knocked into the endogenous Drak locus, Drak-GFP co-localized with Filamin during cellularization in early embryos. Drak mutant embryos showed slower cellularization dynamics and reduced phosphorylation of Spaghetti squash (sqh), the regulatory light chain of *Drosophila* myosin II that is functionally required during cellularization.

The authors further show that these three proteins (Drak, Filamin, Sqh) localize to muscle attachment sites in developing indirect flight muscles. While initially expressed in either attaching myotubes or tendon cells, the authors find in addition that Drak and Filamin co-localized weakly in tendon cells upon further maturation of the attachment sites. Mutants of Drak and Filamin in its closed form, including their respective double mutant, caused statistically significant lengthening of adult attachment sites. Comparisons with the Filamin-open mutants are, however, not shown.

The phenotypes appear quite subtle, and the participating proteins predominantly show a diffuse distribution pattern making co-localization analysis (Fig. 6, 7) challenging. In addition, the manuscript still contains avoidable errors and needs editing. I have therefore a few suggestions to improve the manuscript.

Major comments:

1) Figure 2: It is not clear how the Filamin fragments were tagged and purified. According to Methods (page 5), they were purified as described in Huelsmann et al. 2016. However, since the authors also re-name the constructs, they should also briefly describe their purification procedures, and how the purified proteins were treated for the pull-down experiments. As it is currently described, the assay would not work. There is also no reference for the pGTVL1-SCG vector (Features? Company? Reference?). In addition, both binding and washing buffers do not contain any detergent. In the absence of detergent, recombinant proteins tend to interact non-specifically. Furthermore, no inputs controls of the purified proteins are shown. This is clearly necessary to judge the starting material in relation to the recovered complexes. It is common that interactions in yeast two-hybrid screens are confirmed by at least 2 independent biochemical assays. The authors should therefore attempt, for example, to immunoprecipitate Filamin-Drak complexes from embryos or perform related experiments.

2) Figure 3: By inserting GFP into the genomic locus of Drak, the authors generate a new useful tool, Drak-GFP under control of the endogenous promoter. Insertion in the locus was verified by sequencing (Methods, page 6). However, this form of verification is insufficient as the insertion might affect splicing and/or localization/stability of the Drak. The authors should therefore provide more evidence that we are indeed looking at a Drak-GFP fusion and sequence the mRNA to demonstrate on the sequence level that fusion occurs as planned. Could they detect a shift of Drak in Western blots to the expected size due to the addition of GFP? Does insertion of GFP at the C-terminus interfere with the regulatory function of the C-terminal domain, possibly altering kinase activity? The demonstration of a functional Drak fusion protein and the thorough description of its embryonic expression pattern in the context of previous findings would certainly strengthen the paper.

3) Figure 6: In Fig. 6A the authors show an overview of the Drak-GFP expression pattern in a dissected pupal thorax. While Drak-GFP is broadly and strongly expressed, they focus on a region with quite weak expression, a developing muscle attachment site. Why are high expressing cells less interesting? Where is Filamin expressed in these cells and in the thorax at this early stage? Using Filamin-mCherry (Fig. 6D), the authors should show co-stainings with Drak-GFP in overviews and detailed images. Why is there only a single muscle attachment site? I would expect corresponding sites on the contralateral side.

4) Figure 6 and 7: The authors state that Drak and Filamin co-localize in tendon cells. While this might be true at early stages, i.e. initiation of attachment site formation (Fig. 6), in force generating later stages (myotube compaction), however, they are clearly separated (Fig. 7, Movie S1). For example, at 35-37h APF, when Filamin is upregulated in muscles, its subcellular distribution is highly ordered and punctuate, while Drak-GFP shows a diffuse pattern. According to their model, Drak should bind to mechanically activated Filamin. Could the authors demonstrate this at the myotendinous junction using Filamin mutants?

5) Supplemental Figure S1: The higher bands (70 kDa and higher) in Suppl. Fig. S1A-C are neither marked nor explained. More importantly, the concentrations of the Filamin fusion proteins used in the negative controls (GST) are not stated. Since this is an important parameter for judging the

specificity of the binding reaction, and since the concentrations vary in the experimental assays, this should be indicated in the figure, legend or methods section.

Minor points:

- a) Page 3, line 12: "at the" is repeated. Please delete surplus words.
- b) Page 10: The authors report 8 hits in the two-hybrid screen but mention 11 DRak-related hits in Methods. Please clarify.
- c) The result section has too many brief summaries. At least the one on page 12, line 19-23 should be omitted.
- d) Many references are missing in sentences referring to published work, e.g. on page 3, line 31 (role during cellularization); page 12, line 15-16 (previously reported); page 15, line 9-13 (could not reproduce).
- e) Page 13, line 25-28 should be moved to the beginning of the result section to better explain the point mutations used in the screen.
- f) Figure 1B is neither referred to or explained in the Introduction or Results.
- g) The vector "pHD-sfGFPScarlesDSred" seems to be spelled wrong, at least "DSred" or "Scarles" (page 6, line 11). The authors should double-check this.
- h) Page 18, line 49: Suppl. Fig. S1 should be Suppl. Fig. S2.
- i) Legend to Figure 5: Please replace "defected".
- j) Figure 6 and 7 could be merged, as they show similar tissues just at two different stages.

Reviewer 3

Comments for the author

Korkiamäki reports the novel finding of a mechanically regulated interaction between Filamin and the kinase Drak. The Reviewer is very impressed by the depth of their study and the overall quality.

The work is of certain interest to the fields of mechanobiology and developmental biology. The authors have generated useful new tools and present some compelling biochemical and genetic data.

However, the Reviewer believes that the manuscript can be further improved, since the current version opens more questions and gaps than it answers. Also formally it seems not well composed, nor refined or appears complete. The main conclusions are not robustly supported by the evidence. The primary flaw is a critical negative result from the embryonic cellularization experiments that directly contradicts the authors' central hypothesis of a conserved, linear pathway. The authors fail to adequately address or interpret this contradiction. Furthermore, their interpretation of the supportive genetic data from muscle development is an oversimplification of a complex interaction that likely points to a different biological model (e.g., parallel pathways) than the one proposed.

Thus, my suggestion is to revise the manuscript in a more concise and logical manner so that it become more tight and robust. Toning down the claims and expose existing observation and knowledge gap will not harm its merit and in the opposite. The manuscript would be significantly strengthened by acknowledging and reconcile the complexity and discussing this alternative model, which leads to a different conclusion about the functional relationship between the two proteins.

1. A complete re-framing of the manuscript to address the context-dependent nature of the Filamin-Drak interaction, acknowledging that the proposed pathway is not supported in the cellularization context.
2. A much more thorough and critical interpretation of the genetic epistasis data.
3. Crucially, new experimental data, in the form of an in vitro kinase assay, is needed to demonstrate that Filamin binding actually regulates Drak's catalytic activity. Without this, the proposed mechanotransduction pathway remains a correlation.
4. The C-terminal region of Drak, where the interaction was mapped, is predicted to be intrinsically disordered, containing a high proportion of polar and charged residues. This characteristic makes it a candidate for non-specific interactions within the artificial environment that causing common false-positive outcome in Y2H.
5. Y2H experiment lacks of essential negative controls. A rigorous Y2H-based discovery pipeline would not only screen with the "open" bait but would also perform parallel screens with "wild-type" and constitutively "closed" versions of the Filamin MSR bait. If Drak were a truly specific, mechanically regulated interactor, it should appear as a hit with the "open" bait but be absent or significantly depleted from the hit lists of the "wild-type" and "closed" baits.
6. the manuscript provides no information on the identity of the other 238 hits from the screen. Without this context, it is impossible to determine if Drak is a uniquely specific interactor for the open Filamin MSR or if it is simply one member of a broader class of proteins that can non-specifically bind to the engineered bait construct.
7. This specific quantitative signature of high co-occurrence but only moderate correlation argues against a simple model where Filamin and Drak form a stable, one-to-one complex. Instead, it is more consistent with a model where Filamin, as a major actin cross-linker, forms a broad structural domain, and a smaller, sub-stoichiometric, and possibly transient population of Drak is recruited to this larger domain. In this scenario, Filamin acts as a general scaffold or platform, not as a dedicated activator for every Drak molecule present. This subtle but important distinction is lost in the authors' discussion, which uses the colocalization data to imply a more direct and stoichiometric functional link than the data may actually support. Later in cellularization, Drak becomes mostly diffuse while Filamin remains strongly localized to the contracting ring, further suggesting that their association is transient and context-dependent even within a single developmental process.
8. Fig 5: test the proposed linear mechanotransduction pathway: Force - Filamin-open - Drak binding - Myosin phosphorylation - Ring contraction. A key prediction is that blocking the Filamin-Drak interaction should phenocopy a Drak depletion. The Filamin-closed MSR mutant was designed to do precisely this by stabilizing the autoinhibited conformation, preventing Drak binding. Contrary to this prediction, Drak-KO embryos display the expected defects actomyosin rings remain hexagonal and pMyosin levels are reduced, confirming assay validity. However, Filamin-closed mutants are indistinguishable from wild type in both ring circularity and pMyosin, directly contradicting the model. The authors' suggestion that the mutant retains partial function is ad hoc and unsubstantiated, and citing severe Filamin deletions does not resolve the clear null phenotype observed.

Alternative explanations merit consideration. Forces during early cellularization may be insufficient to open Filamin, rendering the mechanosensory pathway inactive, with Drak activated independently. Redundant kinase pathways, including Rho-kinase, likely drive most myosin phosphorylation, relegating the Filamin-Drak interaction to a minor or modulatory role. Finally, Filamin's structural actin-crosslinking function may be primary, preserved in the mutant, while mechanosensory signaling is non-essential for cellularization. Overall, colocalization is transient and sub-stoichiometric, and the functional genetics contradict the model's core prediction.

Current version does not adequately address this discrepancy. The most parsimonious interpretation is that Filamin-Drak mechanosensing is not the primary driver of Drak-dependent myosin phosphorylation during embryonic cellularization. The discussion could be strengthened by

explicitly acknowledging this limitation and reframing the Filamin-Drak pathway as a potentially minor or context-dependent contributor rather than the dominant mechanism.

Reviewer's Responses to Questions

Experimental quality

Does each figure have the proper controls?

If 'No', please indicate reasons in Comments for Author box below.

Reviewer #1:

- No

Reviewer #2:

- No

Reviewer #3:

- Yes

Were the data analyzed using appropriate statistical tests?

If 'No', please indicate reasons in Comments for Author box below.

Reviewer #1:

- No

Reviewer #2:

- Yes

Reviewer #3:

- Yes

Reproducibility

Were experiments performed using adequate number of biological replicates?

If 'No', please indicate reasons in Comments for Author box below.

Reviewer #1:

- No

Reviewer #2:

- Yes

Reviewer #3:

- Yes

Does the methods section provide sufficient detail to permit reproducibility?

If 'No', please indicate reasons in Comments for Author box below.

Reviewer #1:

- Yes

Reviewer #2:

- No

Reviewer #3:

- Yes

Completeness

Are the manuscript's conclusions supported by the data?

If 'No', please indicate reasons in Comments for Author box below.

Reviewer #1:

- No

Reviewer #2:

- No

Reviewer #3:

- Yes

Scholarship

Do the authors cite and discuss the merits of data that would argue for and against their conclusion?

If 'No', please indicate reasons in Comments for Author box below.

Reviewer #1:

- Yes

Reviewer #2:

- No

Reviewer #3:

- Yes

Does the manuscript title & abstract accurately reflect the contents of the manuscript, without hyperbole?

If 'No', please indicate reasons in Comments for Author box below.

Reviewer #1:

- Yes

Reviewer #2:

- No

Reviewer #3:

- Yes

First revision

Author response to reviewers' comments

We thank the reviewers for the detailed and expert comments and suggestions. We have tried to correct all the errors and mistakes noted by the reviewers and tuned down our claims about the mechanotransduction pathway. However, we feel that the number of further experiments suggested by the reviewers was all too large to be included in a single report. Many of the suggested experiments such as the full description of Drak-GFP expression in developing and adult *Drosophila* tissues or biochemical characterization of its kinase activity are possible themes of future research report, but in our opinion, outside the focus of this manuscript.

The main changes to the manuscript are.

-We have tuned down the language about the interpretation of our results and changed the title of the manuscript from “Drak is a potential mechanotransduction partner of Filamin” to “Drak is a potential binding partner of *Drosophila* Filamin”

-We have included all yeast two hybrid screen results in the supplemental material (Table S1)

-We have added p-values for Pearson correlation of the co-localization analysis (main text and

Table S2 and S3)

-We now explain more clearly the use of Spaghetti Squash-mCherry and Filamin-mCherry as markers of the developing myotubes and tendon cells, respectively, during the time-lapse analysis of IFM development in pupal stage (current Fig. 6). To make this clear to the reader, we changed the order of figures 6 and 7.

-We now also show the the autofluorescent signal during pupa imaging (Fig. 7)

-We now discuss the limitations of the Drak-GFP allele, including the failure to detect with anti-GFP by immunoblotting (new Fig. S5)

We feel that in the current form our manuscript contains valid experimental data about the in vitro interaction between Drosophila Filamin and Drosophila Drak. In addition, we report a moderate but transient colocalization in two developmental processes between Drak-GFP and Filamin-mCherry genomic fusion protein products. Even if we could not fully confirm the Drak - Filamin interaction in vivo, our data provide new information about Drosophila Drak expression and localization. We hope that the manuscript is suitable for publication in *Biology Open* in its current form.

Response to reviewer's comments (January 2026)

Reviewer 1:

require clarification or strengthening before publication. In particular, the authors should (i) address the apparent discrepancy between the in vitro binding assays and the in vivo phenotypic effects of the Filamin-closed mutant during cellularization, and (ii) provide a more clear interpretation of Drak's functional contribution in muscle development, given the largely diffuse localization at later pupal stages. Some concerns are listed below:

Authors' response

i. In the previous version of the manuscript, we stated: "Interaction of Drak with open Filamin MSR mutation and transient localization at priming phase suggest that Filamin might be involved in Drak function at the forming cellularization furrow. Unfortunately, we could not confirm the role of Filamin in Drak function by using the Filamin closed MSR mutation in vivo... It is possible that our Filamin closed MSR mutant is still partially functional in vivo." In the current, revised version now further discuss the possibility that Filamin closed MSR mutant is partially functional and cited to previous studies that this mutant is expected to require approximately 10 pN forces for opening, instead of 3,9 pN for the wild type. We note that forces over 10 pN have been observed in vivo.

ii. We have added discussion about the possibility that some of the diffused GFP signal can be caused by GFP-containing Drak degradation product. Our interpretation is that Drak has a minor contribution to the attachment site, possibly at the stage of maturation.

Reviewer 1:

Major comments:

1. Study lacks a potential mechanism by which Filamin regulates Drak kinase activity in response to mechanotransduction.

Authors' response:

In general, there is no data available about the regulation of drak-family kinase activity. There are two major possibilities: 1. Drak is regulated by phosphorylation via other kinases 2. Drak is regulated via interaction with other proteins. We now discuss these alternatives in the end of the discussion section.

Reviewer 1:

2. Authors show that Drak and Filamin colocalize during furrow initiation. However, the Filamin closed MSR mutant does not phenocopy loss of Drak. This weakens the argument that Filamin is essential for Drak recruitment or activity in embryos. Authors should consider redundancy or partial function of the closed-MSR mutant. Along those lines in figure 5, graph in figure 5b doesn't reflect the data in figure 5a with regards to the circularity differences seen between Drak KO vs WT and Filamin mutant. Given that the amount of total myosin is variable between samples, loading control should be included and amount of pMyosin should be represented as ratio of pMyo/total myosin.

Authors' response:

Please, see our answer to the general comment above about the possibility that closed-MSR mutant may be partially functional. We are sorry, there was a labeling error in Figure 5b. The circularity indexes of WT filamin embryos and Filamin-closed embryos are indistinguishable and the Drak-KO embryos deviate from this. This labeling error is now corrected. We have now also added a bar graph showing the change in the ratio of pMyosin/total Myosin fold changes. The use of loading controls was already explained in the previous version and supplemental blot transparency figure (Fig S2) show the total protein staining of all gels used for the phosphomyosin immunoblotting. The added quantification bar graph takes the loading controls in account.

Reviewer 1:

3. In figures 6 and 7, authors suggest that Drak and Filamin co-localize at tendons. However, tendons and myotubes should be stained for using established markers.

Authors' response:

We have now explained that Spaghetti Squash has been used as a marker for myotubes and Filamin for tendon cells. The order of the figures has been changed so that these markers are introduced first.

Reviewer 1:

4. In figure 8, the phenotypic readout (length of attachment sites) is relatively coarse. It would strengthen the argument if additional readouts (e.g. sarcomere integrity) were included to assess the physiological consequences.

Authors' response:

We used the attachment site length as this was used in our previous analysis of attachment site protein mutants. We do not detect changes in sarcomere integrity by Drak knockout.

Reviewer 1:

5. Throughout the paper, authors rely heavily on colocalization analyses using Manders' coefficients and Pearson correlations. While helpful, the sample sizes are limited, and error/variance estimates are sometimes broad. More replicates, and statistical details (e.g. exact p-values, n per image per embryo/pupa) should be presented to solidify conclusions.

Authors' response:

The image and sample amounts were already given in the text for cellularization colocalization analysis but we previously did not mention in the main text the amount of images used for the IFM colocalization analysis. Both are now stated in results text. Already in the previously submitted version, full details of the images and samples

used for the analysis were given in the supplemental tables (table S2 and S3 in the current version). P-values for colocalization parameters are now given in the results. We did one-sample t-tests for each parameter using sample mean values and now report the population mean values with 95% confidence intervals.

Reviewer 2:

Major points

1) Figure 2: It is not clear how the Filamin fragments were tagged and purified. According to Methods (page 5), they were purified as described in Huelsmann et al. 2016. However, since the authors also re-name the constructs, they should also briefly describe their purification procedures, and how the purified proteins were treated for the pull-down experiments. As it is currently described, the assay would not work. There is also no reference for the pGTVL1-SCG vector (Features? Company? Reference?). In addition, both binding and washing buffers do not contain any detergent. In the absence of detergent, recombinant proteins tend to interact non-specifically. Furthermore, no input controls of the purified proteins are shown. This is clearly necessary to judge the starting material in relation to the recovered complexes. It is common that interactions in yeast two-hybrid screens are confirmed by at least 2 independent biochemical assays. The authors should therefore attempt, for example, to immunoprecipitate Filamin-Drak complexes from embryos or perform related experiments.

Authors' response:

In the revised methods section, we have included the specific details about the expression and purification of Drosophila Filamin fragments. To maintain physiological conditions for soluble cytosolic protein-protein interaction, we did not use detergents in our binding and wash buffer. Our binding buffer contains 100 mM NaCl and 1 mM DTT which makes its ionic strength and reducing environment similar to that of the cytosol. The presence of DTT prevents aggregation of protein via non-native disulfide bond formation, and 100 mM salt prevents non-specific interactions that might occur in low-salt concentration. Our lysis and wash buffer (PBS) contains 135 mM of NaCl to ensure that non-specific protein-protein interactions do not occur. This condition ensures specific protein-protein interactions which are biologically relevant. We agree with the reviewer that input controls are necessary to judge the starting material in relation to the recovered complex, and we have included the SDS-PAGE gel image with the varying concentration of purified Drosophila Filamin fragments (Fig S1 panel A)

Reviewer 2:

2) Figure 3: By inserting GFP into the genomic locus of Drak, the authors generate a new useful tool, Drak-GFP under control of the endogenous promoter. Insertion in the locus was verified by sequencing (Methods, page 6). However, this form of verification is insufficient as the insertion might affect splicing and/or localization/stability of the Drak. The authors should therefore provide more evidence that we are indeed looking at a Drak-GFP fusion and sequence the mRNA to demonstrate on the sequence level that fusion occurs as planned. Could they detect a shift of Drak in Western blots to the expected size due to the addition of GFP? Does insertion of GFP at the C-terminus interfere with the regulatory function of the C-terminal domain, possibly altering kinase activity? The demonstration of a functional Drak fusion protein and the thorough description of its embryonic expression pattern in the context of previous findings would certainly strengthen the paper.

Authors' response

We agree with the reviewer that there is a possibility that at least part of the diffuse GFP signal in the Drak-GFP flies can be caused by Drak degradation product containing GFP. Unfortunately, we failed to detect either full length Drak-GFP or GFP alone in anti-GFP immunoblotting, even though the control immunoblots were positive and sensitive at ng level. At this point we can only conclude that Drak-GFP expression level is low. Our analysis of the Drak-GFP signal in the developing IFM suggests anyway that the GFP-signal

follows the expected Drak- expression time course detected at mRNA level.

It would be interesting to study the expression of Drak-GFP in various tissues more closely, but we think that it is not in the scope of this manuscript with focuses on potential interaction with Filamin.

Reviewer 2:

3) Figure 6: In Fig. 6A the authors show an overview of the Drak-GFP expression pattern in a dissected pupal thorax. While Drak-GFP is broadly and strongly expressed, they focus on a region with quite weak expression, a developing muscle attachment site. Why are high expressing cells less interesting? Where is Filamin expressed in these cells and in the thorax at this early stage? Using Filamin-mCherry (Fig. 6D), the authors should show co- stainings with Drak

Authors' response

Previous figure 6 has now been changed to Figure 7. The current figure also contains control images showing that there is a strong green autofluorescent signal in some of the freely moving cells in the pupa. Part of this autofluorescent signal is overlapping with Filamin signal in some of the mobile cells. Previously Drosophila filamin (Cheerio) has been reported in lamellocytes (Rus et al., 2006, now cited in the manuscript).

Reviewer 2:

4) Figure 6 and 7: The authors state that Drak and Filamin co-localize in tendon cells. While this might be true at early stages, i.e. initiation of attachment site formation (Fig. 6), in force generating later stages (myotube compaction), however, they are clearly separated (Fig. 7, Movie S1). For example, at 35-37h APF, when Filamin is upregulated in muscles, it's subcellular distribution is highly ordered and punctuate, while Drak-GFP shows a diffuse pattern. According to their model, Drak should bind to mechanically activated Filamin. Could the authors demonstrate this at the myotendinous junction using Filamin mutants?

Authors' response

Yes, the reviewer is right: we only detect moderate colocalization between Filamin and Drak transiently in the tendon cells. Currently we are not able to image simultaneously mutant Filamin and Drak in living samples, because we have Filamin-mCherry allele only for the WT Filamin.

Reviewer 2:

5) Supplemental Figure S1: The higher bands (70 kDa in Suppl. Fig. S1A-C are neither marked nor explained. More importantly, the concentrations of the Filamin fusion proteins used in the negative controls (GST) are not stated. Since this is an important parameter for judging the specificity of the binding reaction, and since the concentrations vary in the experimental assays, this should be indicated in the figure, legend or methods section.

Authors' response

The 70 kDa are apparently bacterial proteins that bind to glutathione. We see these bands with all Drak fragments tested, also with those that do not bind filamin. Due to space limitation, these data are not shown in the manuscript, but we give the reviewer here an example of the SDS-PAGE analysis of material used for the pull-down experiment (left image) and corresponding amounts of Filamin fragment bound to these Drak constructs (right image)

As answer for the amount of Filamin captured with GST alone, we show below experiments including GST-only pull-downs with all three Filamin fragments used. Routinely, Filamin fragment binding to GST was tested at the highest used Filamin concentration (typically 10 μ M). We include here an earlier pull-down experiment that shows the background level of binding.

Reviewer 2:

Minor points:

- a) Page 3, line 12: "at the" is repeated. Please delete surplus words.
- b) Page 10: The authors report 8 hits in the two-hybrid screen but mention 11
- c) The result section has too many brief summaries. At least the one on page 12, line 19-23 should be omitted.
- d) Many references are missing in sentences referring to published work, e.g. on page 3, line 31 (role during cellularization); page 12, line 15-16 (previously reported); page 15, line 9-13 (could not reproduce).
- e) Page 13, line 25-28 should be moved to the beginning of the result section to better explain the point mutations used in the screen.
- f) Figure 1B is neither referred to or explained in the Introduction or Results.
- g) The vector "pHD-sfGFPScarlesDSred" seems to be spelled wrong, at least "DSred" or "Scarles" (page 6, line 11). The authors should double-check this.
- h) Page 18, line 49: Suppl. Fig. S1 should be Suppl. Fig. S2.
- i) Legend to Figure 5: Please replace "defected".
- j) Figure 6 and 7 could be merged, as they show similar tissues just at two different stages.

Authors' response:

a, c, d, e, f, g, h, and i) We thank the reviewer for the detailed comments and have corrected the issues b) We are sorry about this misunderstanding. Already in the previous version, we mentioned in the Methods section that 11 of the sequences Y2H hits corresponded to Drak, but 4 of these contained exactly the same sequence, and were thus multiple copies of the same clone. Thus, the total number of independent Drak Y2H hits was 8. This can now be confirmed from the added full Y2H data (Table S1).

The tree fragments coloured in orange in Fig. 2 are those that were confirmed at protein level in the pull-down experiment. This is explained in Figure labeling and in the legend.

j) We agree the figures could have been combined. However, we added more microscopy images of the attachment site maturation stage to better showcase Drak-GFP expression during this stage (Fig. 7 in the revised manuscript), as such, the figures are now too large to combine.

Reviewer 3

1. A complete re-framing of the manuscript to address the context-dependent nature of the Filamin-Drak interaction, acknowledging that the proposed pathway is not supported in the cellularization context

Authors' response:

This comment is similar as those from reviewer 1. Major re-framing has been explained above.

Reviewer 3:

2. A much more thorough and critical interpretation of the genetic epistasis data

Authors' response:

We have removed the speculation in the discussion that Filamin may have a role in myosin regulation during IFM development. Instead, we discuss the possible ways how Drak activity may be regulated. (see comment to reviewer 1)

Reviewer 3:

3. Crucially, new experimental data, in the form of an in vitro kinase assay, is needed to demonstrate that Filamin binding actually regulates Drak's catalytic activity. Without this, the proposed mechanotransduction pathway remains a correlation.

Authors' response

As mentioned above, we have tuned down the discussion about the possible mechanotransduction pathway. Instead, we discuss the possible ways how Drak activity could be regulated. Unfortunately, we are not able to express and purify full-length Drak and thus we have not been able to perform in vitro kinase assays.

Reviewer 3:

4. The C-terminal region of Drak where the interaction was mapped, is predicted to be intrinsically disordered, containing a high proportion of polar and charged residues. This characteristic makes it a candidate for non-specific interactions within the artificial environment that causing common false-positive outcome in Y2H.

5. Y2H experiment lacks of essential negative controls. A rigorous Y2H-based discovery pipeline would not only screen with the "open" bait but would also perform parallel screens with "wild-type" and constitutively "closed" versions of the Filamin MSR bait. If Drak were a truly specific, mechanically regulated interactor, it should appear as a hit with the "open" bait but be absent or significantly depleted from the hit lists of the "wild-type" and "closed" baits.

Authors' response

4 and 5) We appreciate the concern regarding non-specific interaction mediated by intrinsically disordered C-terminal region of Drak in Y2H. Furthermore, the use of mutant Filamin in the interaction adds to the concern that the interaction may be an artefact. We chose in vitro pull-down experiment to confirm the Y2H results and our main arguments for the specificity of the interaction are: 1. We could not observe the interaction with all Drak fragments tested, but there appears to be some specificity. 2. Even though open-Filamin showed the best interaction, we also detected interaction with WT Filamin. On the other hand, no interaction was detected with closed-Filamin. Again, this gave some evidence of specificity.

Reviewer 3:

6. the manuscript provides no information on the identity of the other 238 hits from the screen. Without this context, it is impossible to determine if Drak is a uniquely specific interactor for the open Filamin MSR or if it is simply one member of a broader class of proteins that can non-specifically bind to the engineered bait construct.

Authors' response

In the revised version, we have added text regarding the result of the Y2H assay and lists of the hits are presented in Supplementary Table 1.

Reviewer 3:

7. This specific quantitative signature of high co-occurrence but only moderate correlation argues against a simple model where Filamin and Drak form a stable, one-to-one complex. Instead, it is more consistent with a model where Filamin, as a major actin cross-linker, forms a broad structural domain, and a smaller, sub-stoichiometric, and possibly transient population of Drak is recruited to this larger domain. In this scenario, Filamin acts as a general scaffold or platform, not as a dedicated activator for every Drak molecule present.

Authors' response

We largely agree with the reviewer about this interpretation. We have now added more text along these lines at the end of discussion: "This, together with the colocalization patterns that are evident in only specific developmental sub-processes, the largely distinct phenotypes of single mutants but synergistic phenotypes observed in certain contexts, suggests that Filamin is likely not a major regulator of Drak in non-muscle

myosin activation. Instead, the function of the possible interaction is likely context-dependent and operates in short developmental time frames. One possibility is that Filamin recruits or scaffolds Drak at specific subcellular sites, enabling localized signalling that is important in particular processes."

Reviewer 3:

8. Fig 5: test the proposed linear mechanotransduction pathway: Force - Filamin-open - Drak binding - Myosin phosphorylation - Ring contraction. A key prediction is that blocking the Filamin-Drak interaction should phenocopy a Drak depletion. The Filamin-closed MSR mutant was designed to do precisely this by stabilizing the autoinhibited conformation, preventing Drak binding. Contrary to this prediction, Drak-KO embryos display the expected defects actomyosin rings remain hexagonal and pMyosin. Alternative explanations merit consideration. Forces during early cellularization may be insufficient to open Filamin, rendering the mechanosensory pathway inactive, with Drak activated independently. Redundant kinase pathways, including Rho-kinase, likely drive most myosin phosphorylation

Current version does not adequately address this discrepancy. The most parsimonious interpretation is that Filamin-Drak mechanosensing is not the primary driver of Drak-dependent myosin phosphorylation during embryonic cellularization. The discussion could be strengthened by explicitly acknowledging this limitation and reframing the Filamin-Drak pathway as a potentially minor or context-dependent contributor rather than the dominant mechanism.

Authors' response:

Please see the previous response to comment 7). We have now included this view.

Second decision letter

MS ID#: bio.062185R1

MS Title: Drak is a potential binding partner of Drosophila Filamin

Authors: Riku Oskari Korkiamäki; Chandan Thapa; Jari P Yläanne; Hannah J Green

I have now reached a decision on the above manuscript.

The reviewer reports are shown at the bottom of this email.

As you will see, the reviewers gave favourable reports, but Reviewer 2 raised some final points that will require amendments to your manuscript. Please address the point regarding "autofluorescence", as well as the minor point raised. I hope that you will be able to address these comments, because we would like to be able to accept your paper.

At this stage, we also ask you to ensure your manuscript complies with our formatting guidelines - please see our manuscript preparation guidelines for details. Provided you are able to fully address the referees' comments, we are positive about publication of your paper (we accept over 95% of revision submissions) and therefore hope you won't mind any extra work involved in reformatting your manuscript at this point.

Please upload both a 'clean' version of your Word file, along with a highlighted version clearly showing where you have made changes in the revised manuscript. Please avoid using 'Track changes' in Word files as these are lost in PDF conversion.

I should be grateful if you would also provide a point-by-point response detailing how you have dealt with the points raised by the reviewers in the 'Response to Reviewers' box. Please attend to

all of the reviewers' comments. If you do not agree with any of their criticisms or suggestions please explain clearly why this is so.

Reviewer 2

Comments for the author

In their revised manuscript, Korkiamäki et al. provide new data for their finding that "Drak is a potential binding partner of *Drosophila* Filamin". In particular, they add more methodological information for the expression and purification of the *Drosophila* filamin fragments and the buffers they used for the in vitro pull-down assays. These important experimental details are now included in the revised method section. In addition, they also convincingly show that unspecific binding of any of the filamin fragments to GST was tested at the highest concentrations used for the experiments using GST-Drak.

Other points were less well addressed and remain unresolved. These are an independent biochemical confirmation (other than GST-pull-downs) of the interactions between Filamin and Drak as it is common for interactions coming out of yeast two-hybrid screens. The integration of GFP into the Drak locus was tested only on the genomic level and not on the mRNA level. In addition, co-localization at attachment sites is not convincingly shown (Fig. 7). The question remains if the "autofluorescence" detected at 488 nm (Fig. 7) is indeed autofluorescence or a real signal stemming from Drak-GFP expressed in hemocytes. Usually, autofluorescence (e.g. from cuticular structures) is detected in several channels and not only in a single channel. I suggest to examine the co-localization also at later stages, when attachment sites have formed.

One minor point concerns the legend to Suppl. Figures S1: "In the first right image of (A) panel..." should read "In the first right images of (B) panel".

Reviewer 3: Very nice work with great revision.

Reviewer's Responses to Questions

Experimental quality

Does each figure have the proper controls?

If 'No', please indicate reasons in Comments for Author box below.

Reviewer #2:

- No

Reviewer #3:

- Yes

Were the data analyzed using appropriate statistical tests?

If 'No', please indicate reasons in Comments for Author box below.

Reviewer #2:

- Yes

Reviewer #3:

- Yes

Reproducibility

Were experiments performed using adequate number of biological replicates?

If 'No', please indicate reasons in Comments for Author box below.

Reviewer #2:

- No

Reviewer #3:

- Yes

Does the methods section provide sufficient detail to permit reproducibility?

If 'No', please indicate reasons in Comments for Author box below.

Reviewer #2:

- Yes

Reviewer #3:

- Yes

Completeness

Are the manuscript's conclusions supported by the data?

If 'No', please indicate reasons in Comments for Author box below.

Reviewer #2:

- No

Reviewer #3:

- Yes

Scholarship

Do the authors cite and discuss the merits of data that would argue for and against their conclusion?

If 'No', please indicate reasons in Comments for Author box below.

Reviewer #2:

- Yes

Reviewer #3:

- Yes

Does the manuscript title & abstract accurately reflect the contents of the manuscript, without hyperbole?

If 'No', please indicate reasons in Comments for Author box below.

Reviewer #2:

- Yes

Reviewer #3:

- Yes

We thank the reviewers for the comments. We have addressed concerns regarding autofluorescence during pupal flight muscle development.

The main changes to the manuscript are:

-We changed the labelling of panels showing autofluorescence and we described the experiment setup regarding autofluorescence in both figures 3 and 7 with more detail in the corresponding results text and figure legends.

-We added an explanation to the results text on how we distinguished the autofluorescent signal stemming from free-moving cells and tendon-specific signal during pupal development.

-We added "Data availability" section with a DOI to the confocal dataset and the associated analysis code used in this manuscript.

-We edited figure labelling font and size according to the guidelines of Biology Open Journal.

We hope that the manuscript is suitable for publication in *Biology Open* in its current form.

Response to reviewer's comments

Reviewer 2:

In their revised manuscript, Korkiamäki et al. provide new data for their finding that "Drak is a potential binding partner of *Drosophila* Filamin". In particular, they add more methodological information for the expression and purification of the *Drosophila* filamin fragments and the buffers they used for the in vitro pull-down assays. These important experimental details are now included in the revised method section. In addition, they also convincingly show that unspecific binding of any of the filamin fragments to GST was tested at the highest concentrations used for the experiments using GST-Drak.

Other points were less well addressed and remain unresolved. These are an independent biochemical confirmation (other than GST-pull-downs) of the interactions between Filamin and Drak as it is common for interactions coming out of yeast two-hybrid screens. The integration of GFP into the Drak locus was tested only on the genomic level and not on the mRNA level. In addition, co-localization at attachment sites is not convincingly shown (Fig. 7). The question remains if the "autofluorescence" detected at 488 nm (Fig. 7) is indeed autofluorescence or a real signal stemming from Drak-GFP expressed in hemocytes. Usually, autofluorescence (e.g. from cuticular structures) is detected in several channels and not only in a single channel. I suggest to examine the co-localization also at later stages, when attachment sites have formed.

One minor point concerns the legend to Suppl. Figures S1: "In the first right image of (A) panel..." should read "In the first right images of (B) panel".

Authors' response

We think that the autofluorescence question was a misunderstanding by the reviewer. The imaged fly, marked as "488 autofl." in Fig. 7B, did not contain any Drak-GFP, as the fly had only the Filamin-mCherry allele. It was imaged and treated with identical imaging parameters and brightness/contrast controls as the flies with both Drak-GFP and Filamin-mCherry alleles. Similarly, Filamin-mCherry autofluorescence and background were studied by imaging flies without the Filamin-mCherry allele. To avoid similar misunderstanding by readers, in the revised manuscript, we clarified this by renaming the panels as "no Drak-GFP" and "no Filamin-mCherry" to show the corresponding autofluorescence and background signal in those channels. Additionally, we added explanation to the results-section that addresses the free-moving cells (we do not specify further which cells they might be). We state that the signal stemming from free-moving cells was typically brighter and round, whereas the tendon-associated signal followed the Filamin-mCherry-marked attachment site structure and it appeared at the expected time and location along Filamin-mCherry, with consistent signal intensity and morphology across several pupae.

Regarding the comment of testing Drak and Filamin colocalization in mature attachment sites, we tested it at the compaction stage, but we saw no colocalization either in tendon or muscle. We state in manuscript: "During the compaction stage, Drak-GFP distribution was diffuse and no structural concentration to attachment sites or filamentous structures was observed (Fig. 6B)". In the adult muscle, Drak-GFP level is too low for colocalization analysis.

The error in supplementary Figure S1 has been corrected.

Third decision letter

MS ID#: bio.062185R2

MS Title: Drak is a potential binding partner of Drosophila Filamin

Authors: Riku Oskari Korkiamäki; Chandan Thapa; Jari P Yläanne; Hannah J Green

I am happy to tell you that your manuscript has been accepted for publication in Biology Open, pending our standard publication integrity checks. It was accepted on 24th March 2026.